# DiffHammer: Rethinking the Robustness of Diffusion-Based Adversarial Purification

**Kaibo Wang**[1], **Xiaowen Fu**[1], **Yuxuan Han**[1], **Yang Xiang**[1,2]*
[1]Department of Mathematics, The Hong Kong University of Science and Technology
[2]HKUST Shenzhen-Hong Kong Collaborative Innovation Research Institute
{kwangbi, xfuak, yhanat}@connect.ust.hk, maxiang@ust.hk

## Abstract

Diffusion-based purification has demonstrated impressive robustness as an adversarial defense. However, concerns exist about whether this robustness arises from insufficient evaluation. Our research shows that EOT-based attacks face gradient dilemmas due to global gradient averaging, resulting in ineffective evaluations. Additionally, 1-evaluation underestimates resubmit risks in stochastic defenses. To address these issues, we propose an effective and efficient attack named DiffHammer. This method bypasses the gradient dilemma through selective attacks on vulnerable purifications, incorporating $N$-evaluation into loops and using gradient grafting for comprehensive and efficient evaluations. Our experiments validate that DiffHammer achieves effective results within 10-30 iterations, outperforming other methods. This calls into question the reliability of diffusion-based purification after mitigating the gradient dilemma and scrutinizing its resubmit risk.

## 1 Introduction

The vulnerability of deep neural networks (DNNs) to adversarial samples hinders their application in security-critical domains, where attackers can deceive DNNs by introducing carefully crafted noises [25, 9]. To mitigate this issue, numerous defense strategies have been proposed to enhance their robustness, among which *diffusion-based purification* has emerged as a promising approach [22, 26, 3]. Diffusion models [11, 24] are designed to construct stochastic processes from noisy data distributions to cleaner ones. As a result, the presence of small adversarial noise can be drowned out by larger noise, which can then be iteratively denoised using the diffusion model [22, 26] or purified through a diffusion-involved optimization [3]. The iterative algorithm and stochasticity of diffusion enhance their purification capabilities empirically, yet they also present a challenge in evaluating their robustness [16, 13]. There are ongoing concerns regarding their effectiveness:

*Whether their effectiveness originated from inherent robustness or insufficient evaluation?*

The Expectation of Transformation (EOT) [2] method allows attacks to adapt across stochastic purifications by averaging the gradients of sampled purifications. It maximizes attack success rate under the assumption that most purifications share a common vulnerability, i.e., susceptible to a same adversarial noise. However, high stochasticity in diffusion-based purification challenges this assumption, rendering EOT-based attacks ineffective. Purifications with unshared vulnerability ($\mathcal{S}_0$, shown in Figure 1) will provide inconsistent gradients in the attack, leading to a *gradient dilemma*. Moreover, obtaining gradients for purification is time-consuming, and this dilemma further increases computational overhead.

*Corresponding author

38th Conference on Neural Information Processing Systems (NeurIPS 2024).

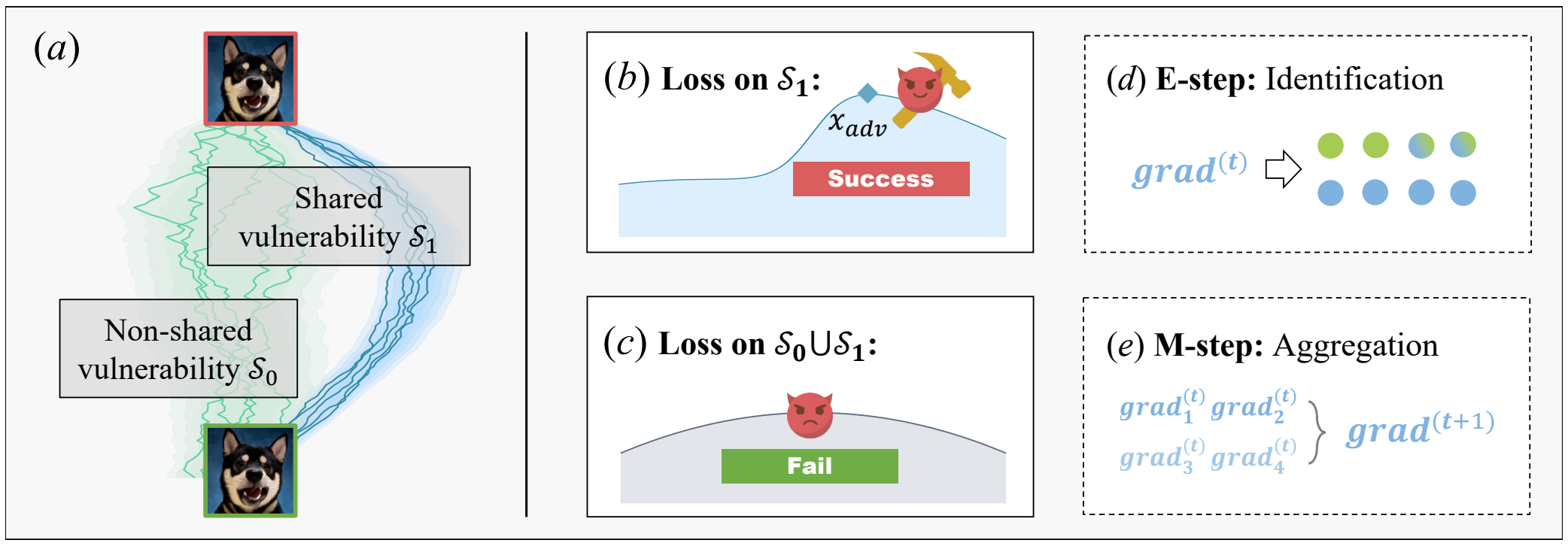

Figure 1: Illustration of DiffHammer. From left to right: (a) Set of purifications with unshared ($\mathcal{S}_0$) / shared ($\mathcal{S}_1$) vulnerabilities. (b,c) Maximizing the loss function on $\mathcal{S}_0 \cup \mathcal{S}_1$ may lead to less effective attacks than on $\mathcal{S}_1$. (d,e) Selective attack targets on $\mathcal{S}_1$ to avoid the gradient dilemma. We identify purifications from $\mathcal{S}_1$ in the E-step and aggregate the gradients of these purifications in the M-step.

To address this, we introduce a selective attack method, *DiffHammer*[2], based on expectation maximization (EM) [5]. As shown in Figure 1, our algorithm bypasses the gradient dilemma by iteratively identifying vulnerable purifications in the E-step and aggregating their gradients in the M-step. This approach focuses on shared vulnerabilities, akin to using a hammer on weak spots rather than the entire structure. Additionally, we enhance attack efficiency by reducing the diffusion process's backpropagation complexity from $\mathcal{O}(N)$ to $\mathcal{O}(1)$ through gradient grafting.

Another issue with diffusion-based purification is insufficient evaluation. Attackers aiming for a single success, such as logging in, can achieve a higher chance of success by resubmitting an adversarial sample to the stochastic defense [20]. This resubmit risk is underestimated in 1-evaluation tailored for deterministic defenses, especially given the high stochasticity of the diffusion process. To address this, we upgraded 1-evaluation to $N$-evaluation and seamlessly integrated it into our DiffHammer framework, offering a more comprehensive risk assessment without extra time costs.

We revisit the robustness of diffusion-based purification within the DiffHammer framework. DiffHammer surpasses other state-of-the-art attacks [4, 21, 13, 16, 1, 2] by circumventing the gradient dilemma and achieving near-optimal effectiveness within 10-30 iterations. Under $N$-evaluation protocol and our selective attack, we find that the risk associated with diffusion-based purification is significantly underestimated. Mainstream purifications [22, 26, 3] fail to withstand even 10 resubmit attacks and may degrade the performance of robust models, indicating that their robustness potential remains underutilized. We hope that DiffHammer's insights into evaluating diffusion-based purification robustness will foster more robust defenses. Our main contributions are summarized as follows:

- We identified the limitation of EOT-based attacks and proposed an efficient and effective evaluation, termed DiffHammer, to comprehensively diagnose the adversarial risks of diffusion-based purification.
- We introduce a selective attack that avoids the gradient dilemma by targeting the shared vulnerabilities, with the process expedited by gradient grafting.
- We validate the effectiveness of DiffHammer through extensive experiments on mainstream purifications, revealing underestimated risks through standardized $N$-evaluation.

## 2 Preliminary

### 2.1 Adversarial attacks

Given an image $x \in \mathbb{R}^d$ and its label $y \in [K]$, a classifier $f : \mathbb{R}^d \to \mathbb{R}^K$ with a preprocessor $\phi : \mathbb{R}^d \to \mathbb{R}^d$ will classify it as $f[\phi(x)] = \arg\max_{k=1,\dots,K} f_k[\phi(x)]$, where $K$ is the number of classes. The attacker aims to add an imperceptible adversarial noise $r$ on the image to make it misclassified, i.e., $f[\phi(x + r)] \neq y$. Adversarial noise is typically obtained by maximizing the loss

[2]The codes are publicly available at `https://github.com/Ka1b0/DiffHammer`.

function $\mathcal{L}$ while imposing $\ell_p$-norm constraint for imperceptibility. For instance, projected gradient descent (PGD) [21] updates $\|r\|_\infty \leq \epsilon$ bounded adversarial samples as:

$$r^{(t+1)} = \Pi_{\|r\|_\infty \leq \epsilon}(r^{(t)} + \alpha \text{sign} \nabla_x \mathcal{L}(f[\phi(x + r^{(t)})], y)), \tag{1}$$

where $\alpha$ is step size and $\Pi$ indicates a projection operator.

Preprocessors tailored to defend against adversarial noise may lead to vanishing, exploding, or inaccurate gradients. Attackers can resort to approximating the gradient using Backward Pass Differentiable Approximation (BPDA) [1], where the preprocessor is typically treated as an identity mapping during the backpropagation:

$$\nabla_x \mathcal{L}(f[\phi(x + r^{(t)})], y) \mid_{x+r^{(t)}} \approx \nabla_x \mathcal{L}(f[\phi(x + r^{(t)})], y) \mid_{\phi(x+r^{(t)})} . \tag{2}$$

EOT [2] can be utilized as an additional approach to estimate gradients of stochastic preprocessors, whose effectiveness is derived from the assumption that different preprocessors have shared vulnerabilities. Given $N$ sampled preprocessors $\phi_i$, the estimated gradient is:

$$\nabla_x \mathcal{L}(f[\phi(x + r^{(t)})], y) \approx \frac{1}{N} \sum_{i=1}^{N} \nabla_x \mathcal{L}(f[\phi_i(x + r^{(t)})], y). \tag{3}$$

### 2.2 Diffusion-based purification

The diffusion model [11, 24, 14] establishes a link between the clean data distribution $p_0(x)$ and the noisy distribution $p_1(x)$ through forward (noising) and backward (denoising) processes. The forward process $x_t$ can be expressed as a stochastic differential equation (SDE) for $t$ from 0 to 1 [24]:

$$dx = \mathbf{f}(x, t)dt + \mathbf{g}(t)dw, \tag{4}$$

where $x_0 \sim p_0(x)$, $w_t \in \mathbb{R}^d$ is a standard Wiener process, $\mathbf{f} : \mathbb{R}^d \times \mathbb{R}^d \to \mathbb{R}^d$ is the drift coefficient and $\mathbf{g} : \mathbb{R} \to \mathbb{R}$ is the diffusion coefficient which is designed so that $p_1(x)$ follows a standard Gaussian distribution $\mathcal{N}(0, I_d)$. DDPM [11] can be viewed as a special case of $\mathbf{f}(x, t) := -\beta(t)x/2$ and $\mathbf{g}(t) := \sqrt{\beta(t)}$, where $\beta(t)$ is the noise scheduler, usually set as a linear function w.r.t. $t$.

The evolution of reverse-time SDE ($t$ from 1 to 0) then corresponds to the generation of samples:

$$d\hat{x} = [\mathbf{f}(\hat{x}, t) - \mathbf{g}(t)^2 \nabla_{\hat{x}} p_t(\hat{x})]dt + \mathbf{g}(t)d\bar{w}, \tag{5}$$

where $dt$ is an infinitesimal negative time step and $\bar{w}_t$ is a standard reverse-time Wiener process, and $\nabla_{\hat{x}} p_t(\hat{x})$ is known as time-dependent score function and is typically estimated by the neural network.

Well-trained diffusion models possess the ability to accurately model the score function and denoise noisy samples, enabling adversarial purification. Considering the small magnitude of adversarial noise, Diffpure [22] therefore floods the adversarial noise while preserving the semantic information by running a forward process from 0 to $t^*$, and then purifies it through a denoising process. GDMP [26] leverages the distance from the initial samples as guidance to further preserve semantic information. Leveraging the fact that diffusion model approximates the score function of $x \sim p_0(x)$ more accurately, likelihood maximization (LM) [3] optimizes adversarial samples to minimize estimation error, aligning their distribution with $p_0(x)$.

## 3 DiffHammer

To address ineffectiveness in the attack phase and insufficiency in the evaluation phase, we propose DiffHammer to assess the robustness of diffusion-based purification. First, DiffHammer overcomes the gradient dilemma by designing a *selective attack* (Section 3.1) using the EM algorithm. Furthermore, we seamlessly integrate $N$*-evaluation* (Section 3.2) into the algorithm, enabling more accurate resubmit risk diagnosis and facilitating the attack by providing approximate gradients.

### 3.1 Selective attack

#### 3.1.1 EM algorithm

**Notation and objective.** For a sample $x \in \mathbb{R}^d$ with label $y \in [K]$, we aim to design adversarial noise $r \in \mathbb{R}^d$ to mislead a classifier $f$ with a stochastic purification $\phi : \mathbb{R}^d \to \mathbb{R}^d$. Let $\mathcal{A}$ represent

the misclassification event where $f[\phi(x + r)] \neq y$. Our goal is to maximize the probability of misclassification, expressed as $P(\mathcal{A} \mid r) = \mathbb{E}_\phi p(\mathcal{A} \mid r)$. Here $P$ is the probability density associated with the stochastic purification $\phi$, and $p$ denotes the probability density in logits of $f$ given a specific $\phi$. We refer to the gradient of loss $\mathcal{L}(f[\phi(x+r)], y)$ w.r.t. $x$, $\phi(x+r)$ as the *full gradient* ($\nabla_x \mathcal{L}_\phi$) and the *approximate gradient* ($\nabla_{\phi(x+r)} \mathcal{L}_\phi$), respectively. Without ambiguity, we sometimes abbreviate the loss as $\mathcal{L}_\phi$ and gradients as $\nabla_x \mathcal{L}_\phi$, $\nabla_\phi \mathcal{L}_\phi$.

**Assumption.** We assume that diffusion-based purifications have unshared vulnerabilities, dividing $\phi$ into two sets. We denote $\mathcal{S}_1$ as the largest set of $\phi$ with shared vulnerability, i.e., $\mathcal{S}_1 = \arg\max P(\{\phi : f[\phi(x+r^\star)] \neq y \text{ for a same } r^\star\})$ and $\mathcal{S}_0 = \bar{\mathcal{S}}_1$ as the set of $\phi$ compromised by inconsistent adversarial noise $r$. We assume that optimization towards $\phi \in \mathcal{S}_0$ suffers from the *gradient dilemma*, leading to ineffective attacks. Our task is to identify $\mathcal{S}_1$ and design $r$ for $\phi \in \mathcal{S}_1$.

Indicating whether a $\phi$ belongs to $\mathcal{S}_1$ or $\mathcal{S}_0$ by $z = 1$ or 0, we denote $q(z)$ as the estimated distribution for $z$ for given $\phi$. According to the Jensen's inequality, we maximize a lower bound of our goal:

$$\max \mathbb{E}_\phi \ln \sum_{z=0,1} p(\mathcal{A}, z \mid r) = \mathbb{E}_\phi \underbrace{\sum_{z=0,1} q(z) \ln \frac{p(\mathcal{A}, z \mid r)}{q(z)}}_{\mathcal{Q}(q,r)} \underbrace{- \sum_{z=0,1} q(z) \ln \frac{p(z \mid \mathcal{A}, r)}{q(z)}}_{\mathrm{KL}(q \| p(z|\mathcal{A},r))} \tag{6}$$

where $\mathcal{Q}(q, r)$ represents the evidence lower bound, and $\mathrm{KL}(q \parallel p(z \mid \mathcal{A}, r))$ is the KL divergence between $q$ and the posterior distribution of $z$. Given the coupling of variables $r$ and $z$ in the optimization process, we employ the EM algorithm [5] as a solver that alternates between optimizing $r$ (maximizing $\mathcal{Q}(q, r)$) in the *M-step* and estimating $z$ (minimizing KL divergence) in the *E-step*.

**M-step.** During the M-step, we maximize $\mathcal{Q}(q, r)$ w.r.t. $r$ to raise the objective's lower bound while keeping $q(z)$ fixed. By omitting terms unrelated to $r$, i.e., $q(z) \ln q(z)$ and the priors $\ln p(z = 0), \ln p(z = 1)$, the goal simplifies as

$$\max_r \mathcal{Q}(q, r) \Leftrightarrow \max_r q(z = 1) \ln p(\mathcal{A} \mid r, z = 1) + q(z = 0) \ln p(\mathcal{A} \mid r, z = 0) \tag{7}$$

where the second term is further disregarded in the optimization according to our assumption.

In adversarial attacks, the objective $\mathbb{E}_\phi \ln p(\mathcal{A} \mid r, z = 1)$ is typically replaced by maximizing the loss function $\mathcal{L}_\phi(x + r)$, achieved through average gradient-based methods. Consequently, the M-step update can be integrated into existing attack algorithms like PGD [21] and AA [4] as a *plug-in*, with the difference being the reweighting of the gradient from each $\phi$ by $q(z = 1)$. We further employ stepwise-EM [18] to linearly interpolate the current gradient and the previous gradient with weight $t^{-\alpha}$ for online updating, where $t$ is the number of iterations and $\alpha$ is a hyperparameter. This approach intuitively optimizes the adversarial noise towards the shared vulnerability of $\mathcal{S}_1$, thus avoiding the gradient dilemma in $\mathcal{S}_0$.

**E-step.** During the E-step, we update $q(z = 1)$ to $p(z = 1 \mid \mathcal{A}, r)$ for a given $r$, thereby eliminating the KL divergence, which is the gap between the objective and its lower bound. Notice that $\mathcal{S}_1$ is defined as the *largest* set of $\phi$ susceptible to the same adversarial noise $r^\star$, where usually $r^\star \neq r$. We can estimate $p(z = 1 \mid \mathcal{A}, r)$ by approximating $r^\star$ with by-products $\mathcal{L}_\phi(x + r)$ and $\nabla_x \mathcal{L}_\phi\big|_{x+r}$ in the observation of misclassification event $\mathcal{A}$. Denote the difference between $r^\star$ and $r$ as $\Delta r$, the loss $\mathcal{L}_\phi(x + r^\star)$ w.r.t. $r^\star$ can be linearly approximated since $\Delta r$ is not excessively large, with both $r$ and $r^\star$ bounded by $\|\cdot\|_p \leq \epsilon$:

$$\mathcal{L}_\phi(x + r^\star) \approx \mathcal{L}_\phi(x + r) + \Delta r^T \nabla_x \mathcal{L}_\phi\big|_{x+r} \tag{8}$$

Higher loss increases the likelihood of misclassification, so we map $\mathcal{L}_\phi(x + r^\star)$ to $p(\mathcal{A} \mid r^\star)$ using a monotonically increasing function $\sigma : \mathbb{R} \to [0, 1]$. By the definition of $\mathcal{S}_1$, $r^\star$ compromises as many $\phi$ as possible. We find $r^\star$ through the following optimization:

$$r^\star = r + \Delta r = r + \arg\max \mathbb{E}_\phi \sigma(\mathcal{L}_\phi(x + r) + \Delta r^T \nabla_x \mathcal{L}_\phi\big|_{x+r}) \tag{9}$$

After determining $r^\star$, the probability $q(z = 1)$ that $\phi$ belongs to $\mathcal{S}_1$ can be estimated as $\sigma(\mathcal{L}_\phi(x+r^\star))$.

*Remark.* We use the empirical average w.r.t. $N$ instances $\phi_i, i = 1, \ldots, N$ in each iteration to estimate expectations. In the E-step, we optimize $r^\star$ in a *low-cost* manner, as no model is involved. We reweight the gradient by $\sigma(\mathcal{L}_{\phi_i}(x + r^\star))$ in the M-step and then use it in an off-the-shelf attack algorithm to update the adversarial noise $r$. The primary time cost arises from calculating the gradient $\nabla_x \mathcal{L}_{\phi_i}$ due to the complexity of the purification process. In Section 3.1.2, we will describe how to reduce the $\mathcal{O}(N)$ complexity to $\mathcal{O}(1)$ through gradient grafting.

**Algorithm 1:** DiffHammer

**Input :** Data $(x, y)$, classifier $f$ with stochastic purification $\phi$, number of resubmit $M$, off-the-shelf attack algorithm
**Output :** Robustness $Rob$ in $M$ resubmit attacks

Initialize $r^{(0)}$;
**for** $t \leftarrow 1$ **to** $T$ **do**
  *Evaluation for $t-1$ iteration and input for $t$ iteration*;
  Sample $\phi_i, i = 1, \cdots, N$;
  $Rob^{(t-1)}$ = `Eval` $(r^{(t-1)}, M)$ // for evaluation;
  Record $\mathcal{L}_{\phi_i}^{(t)}, \nabla_{\phi_i}^{(t)} \mathcal{L}_{\phi_i}$ // for attack;
  *E-step: identify the set with shared vulnerability*;
  $\Delta\tilde{r}^{(t)} = \arg\max \sum_i \sigma(\mathcal{L}_{\phi_i}^{(t)} + \Delta\tilde{r}^T \nabla_{\phi_i}^{(t)} \mathcal{L}_{\phi_i})$ ;
  $q_i^{(t)} = \sigma(\mathcal{L}_{\phi_i}^{(t)} + \Delta\tilde{r}^T \nabla_{\phi_i}^{(t)} \mathcal{L}_{\phi_i})$ // probability of affiliation;
  *M-step: estimate the full gradients' aggregation*;
  $\tilde{g}^{(t)} = \sum_i q_i^{(t)} \nabla_{\phi_i}^{(t)} \mathcal{L}_{\phi_i} / N$ // aggregation in $\phi$ stage;
  Select $\hat{\phi}$ according to Equation 11 // representative $\phi$;
  $g^{(t)}$ = `Backward` $(\hat{\phi}(x + r^{(t-1)})^T \tilde{g}^{(t)})$ // gradient grafting;
  $g^{(t)} = t^{-\alpha} g^{(t)} + (1 - t^{-\alpha}) g^{(t-1)}$ // stepwise update;
  $r^{(t)}$ = `AttackAlgorithm` $(r^{(t-1)}, g^{(t)})$;
**end**
$Rob^{(T)}$ = `Eval` $(r^{(T)}, M)$ with $\phi_i, i = 1, \cdots, N$;
**return** $Rob = \min(Rob^{(t)}, t = 1, \cdots, T)$

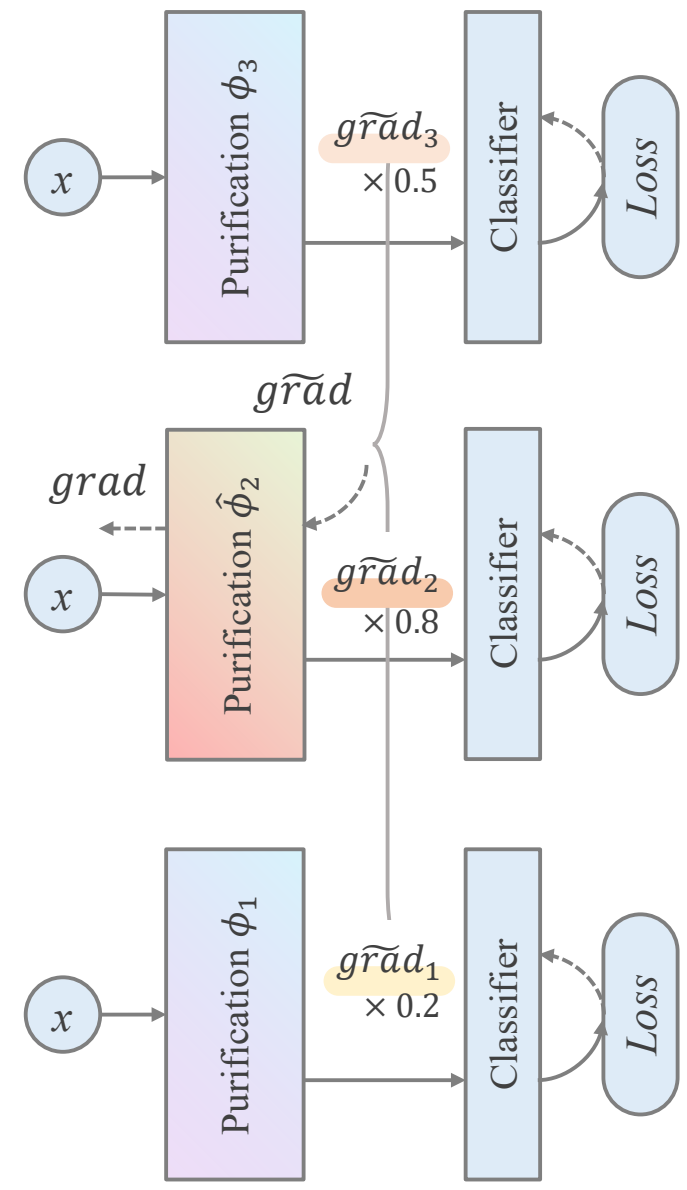

Figure 2: Illustration of efficient gradient aggregation.

#### 3.1.2 Gradient grafting

To address the efficiency challenges in robustness evaluation for diffusion-based purification, we aim to estimate the weighted gradient aggregation $\sum_i q_i(z=1)\nabla_x \mathcal{L}_{\phi_i}$ with minimal computational cost. Our approach aggregates low-cost approximate gradient $\nabla_{\phi_i} \mathcal{L}_{\phi_i}$ in the early stage to estimate the weighted full gradient, which can be expressed as

$$\sum_{i=1}^{N} q_i(z=1)\nabla_x \mathcal{L}_{\phi_i} = \sum_{i=1}^{N} q_i(z=1)\frac{\partial \phi_i}{\partial x}\nabla_{\phi_i}\mathcal{L}_{\phi_i} \approx \frac{\partial \hat{\phi}}{\partial x}\sum_{i=1}^{N} q_i(z=1)\nabla_{\phi_i}\mathcal{L}_{\phi_i}, \tag{10}$$

assuming $\partial\phi_i/\partial x$ can be approximated by $\partial\hat{\phi}/\partial x$. As illustrated in Figure 2, We first compute the weighted approximate gradient $\sum_{i=1}^{N} q_i(z=1)\nabla_{\phi_i}\mathcal{L}_{\phi_i}$, which is then grafted onto $\hat{\phi}$ for backpropagation to estimate the gradient expectation, reducing complexity in backpropagation $x \leftarrow \phi_i(x)$ to $\mathcal{O}(1)$. BPDA [1] uses $I$ to approximate $\partial\phi_i/\partial x$, offering computational simplicity but potentially compromising performance due to oversimplification. Our gradient grafting enhances estimation with one additional backpropagation, achieving a better balance between efficiency and effectiveness. Further related design details are discussed below.

**E-step in $\phi$ stage.** Although $r^\star$ in E-step (Equation 9) involves the full gradient $\nabla_x \mathcal{L}_{\phi_i}$, our main interest is in $\sigma(\mathcal{L}_\phi(x + r^\star))$. Therefore, we can optimize in the stage of $\phi$ rather than $x$ to avoid dependence on the full gradient. We express $\Delta\tilde{r}$ in the stage of $\phi$ as $\Delta\tilde{r} = \phi(x + r^\star) - \phi(x + r)$. The optimization, aimed at *attacking as many $\phi$ as possible*, becomes $\max \mathbb{E}_\phi \sigma(\mathcal{L}_\phi(x+r) + \Delta\tilde{r}^T\nabla_\phi\mathcal{L}_\phi)$, relying only on the approximate gradient. We end up deriving $\sigma(\mathcal{L}_\phi(x + r^\star))$ from optimized $\Delta\tilde{r}$.

**Choice of $\hat{\phi}$.** We aim to select a $\phi$ with representative vulnerability for backpropagation, whose adversarial noise can also affect other $\phi$. For a single-step attack on $\phi$ with adversarial noise $g(\nabla_\phi\mathcal{L}_\phi)$ (e.g. $\text{sign}(\nabla_\phi\mathcal{L}_\phi)$ in $\ell_\infty$ case), we choose $\hat{\phi}$ among $\phi_i$ using the strategy:

$$\hat{\phi} = \underset{\phi \in \{\phi_i, i=1,\cdots,N\}}{\arg\max} \sum_{i=1}^{N} \sigma[\mathcal{L}_{\phi_i}(x+r) + g(\nabla_\phi\mathcal{L}_\phi)^T\nabla_{\phi_i}\mathcal{L}_{\phi_i}]. \tag{11}$$

This consistent optimization goal of *attacking as many $\phi$ as possible* maintains the shared vulnerability from the $\phi$ stage to the $x$ stage. The choice of $\hat{\phi}$ involves discrete optimization within a finite set, solvable by traversal when $N$ is not very large.

### 3.1.3 Discussion

Our selective attack (targeting $\mathcal{S}_1$) enhances both the *effectiveness* and *efficiency* of the EOT-based attack (targeting $\mathcal{S}_0 \cup \mathcal{S}_1$). (1) The EOT-based attack is a specific instance of our algorithm when all $\phi$ share a common vulnerability. In this case, with $\mathcal{S}_0 = \emptyset, q(z=1) = 1$, our selective attack degenerates into an EOT-based attack without side effects. (2) The identification of $\mathcal{S}_1$ involves only approximate gradients, making it nearly cost-free. The grafting trick further increases the efficiency of gradient expectation. (3) The effectiveness of the selective attack arises from avoiding the gradient dilemma, which can cause EOT-based attacks to fail even in simple binary cases.

**Theorem 1** (Failure mode of EOT-based attacks, Proof in Appendix B.1)**.** *Suppose $\phi$ can be divided into two sets, $A$ and $B$. The loss functions w.r.t. $r$ in these sets are defined as $\mathcal{L}_A(r) := \sigma^{-1}(P_{\phi \in A}(\mathcal{A} \mid r))$ and $\mathcal{L}_B(r) := \sigma^{-1}(P_{\phi \in B}(\mathcal{A} \mid r))$, which are $m_A$ and $m_B$ strongly concave, respectively. If the distance between their optimal points $r_A$ and $r_B$ satisfies $\|r_A - r_B\|_2^2 \geq 8 \max\{P(A)\mathcal{L}_A(r_A), P(B)\mathcal{L}_B(r_B)\}/m$ where $m := \min\{P(A)m_A, P(B)m_B\}$, the EOT-based attack is less effective than a simple attack targeting either $A$ or $B$.*

When $r_A$ and $r_B$ are significantly different, $\mathcal{S}_1$ tends to be $A$ or $B$, whereas the other becomes $\mathcal{S}_0$ and provides neutralized gradients. Empirical results indicate that the gradient dilemma is common in diffusion-based purification, highlighting the necessity of selective attacks.

## 3.2 In-loop N-evaluation

Traditional 1-evaluation is insufficient for assessing the robustness of diffusion-based purification, particularly against *resubmit attacks*. As a stochastic defense, the model produces inconsistent results for even the same queries, enabling attackers to achieve desired outcomes through resubmissions. In cases where attack costs are manageable and even a single success is advantageous, e.g., login, defenders should focus on the *model's robustness over $M$ resubmissions*. Thus, we propose using $N$-evaluation as a robustness evaluation protocol for two reasons:

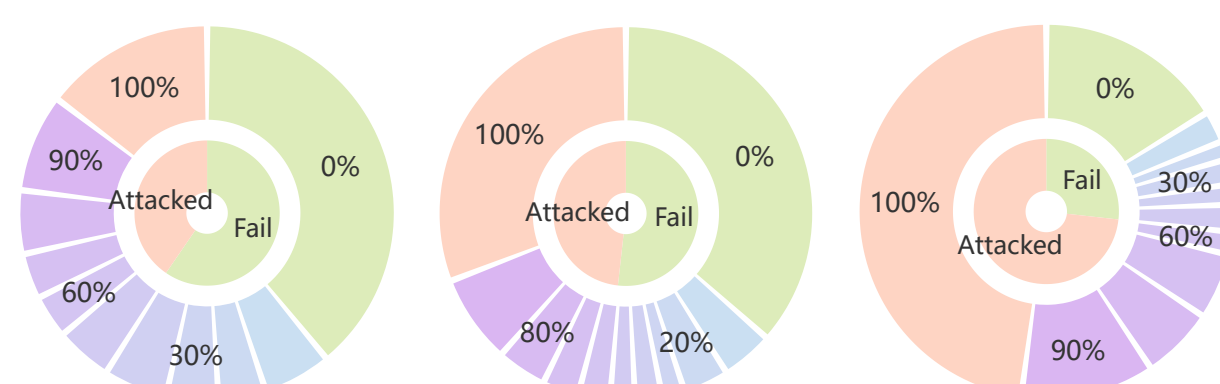

(a) DiffPure [22] (b) GDMP [26] (c) LM [3]

Figure 3: Distribution of attack results ($\ell_\infty : 8/255$) for 1-evaluation (inner ring) and 10-evaluation (outer ring). 32.6%-46.3% of the samples have unshared vulnerabilities, imposing underestimated resubmit risk in 1-evaluation.

(1) As detailed in Theorem 2, 1-evaluation are significantly biased in estimating $M$-resubmit risk. Due to high stochasticity, it is common for $\phi$ to have unshared vulnerabilities, as shown in Figure 3. Merely success or failure record in 1-evaluation fails to capture critical probability information for resubmit risk estimation. For instance, with DiffPure, 46.3% of samples have a attack success rate (ASR) $P(\mathcal{S}_1) \in (0,1)$, leading to a 17.9% overestimation of 10-resubmit robustness in 1-evaluation ($\widehat{Rob}^{(1)}_{MLE} = 41.7\%$, and $Rob = 59.6\%$).

**Theorem 2** (Estimation of the resubmit risk, Proof in Appendix B.2)**.** *Let the sample's robustness in $M$ resubmit attacks $\mathcal{A}_1, \cdots, \mathcal{A}_M$ be denoted as $Rob := P(\mathcal{A}_1 = \cdots = \mathcal{A}_M = 0)$. There exists a uniformly minimum-variance unbiased estimator (UMVUE) for Rob if and only if the number of evaluation trials $N \geq M$. When $N \leq M$, the maximum likelihood estimator (MLE) tends to overestimate Rob in expectation, i.e., $\mathbb{E}(\widehat{Rob}^{(N)}_{MLE}) \geq Rob$.*

(2) $N$-evaluation can be integrated into the attack's loop without extra burden. Samples $\phi_i, i = 1, \cdots, N$ serve both as an evaluation for the previous iteration and input for the current, which will not cause information leakage, as each evaluation involves unseen instances.

Our DiffHammer framework integrates selective attacks and $N$-evaluation, as detailed in Algorithm 1. Notably, our $N$-resubmit robustness extends the traditional 1-submit metric to better assess risks in real-world deployments, orthogonal to our attack algorithm's design. Consequently, DiffHammer offers a more comprehensive risk assessment and enhances attack effectiveness across various metrics.

Table 1: Performance of attacks against diffusion-based purification on CIFAR10. Metrics include Avg./Wor. Rob, (%) and iterations (it.) taken to reach 90% best performance (in parentheses, failure to reach is noted as N/A). Attack algorithms include SOTA white-box attacks with EOT: BPDA [1], PGD [21], AA [4], DA [13], and DiffHammer (DH, ours); and transfer-based attacks (denoted by †): DMI [29], TMI [8], VMI [27], SVRE [30].

| | Defense | DiffPure [22] | | GDMP [26] | | LM [3] | |
|---|---|---|---|---|---|---|---|
| | Metrics | Avg.Rob (it.)↓ | Wor.Rob (it.)↓ | Avg.Rob (it.)↓ | Wor.Rob (it.)↓ | Avg.Rob (it.)↓ | Wor.Rob (it.)↓ |
| | Clean | 90.98 | 76.56 | 93.26 | 83.79 | 87.77 | 74.61 |
| $\ell_\infty : 4/255$ | BPDA | 76.27 (N/A) | 40.82 (126) | 80.61 (N/A) | 52.34 (137) | 69.57 (N/A) | 38.67 (N/A) |
| | DA/AA | 71.52 (N/A) | 40.04 (118) | 73.52 (N/A) | 50.78 (44) | 46.29 (N/A) | 25.78 (N/A) |
| | PGD | 69.80 (125) | 41.02 (114) | 72.32 (N/A) | 50.00 (44) | 40.55 (N/A) | 20.90 (89) |
| | DH | **66.62 (26)** | **35.16 (26)** | **68.36 (18)** | **47.27 (13)** | **29.63 (21)** | **14.45 (16)** |
| $\ell_\infty : 8/255$ | BPDA | 70.74 (N/A) | 36.72 (N/A) | 80.57 (N/A) | 51.95 (N/A) | 55.27 (N/A) | 27.54 (N/A) |
| | DA/AA | 57.60 (N/A) | 33.79 (N/A) | 52.83 (N/A) | 37.70 (N/A) | 32.56 (N/A) | 17.97 (N/A) |
| | PGD | 52.73 (N/A) | 31.05 (112) | 49.41 (N/A) | 36.91 (N/A) | 17.99 (31) | 9.38 (31) |
| | DH | **42.54 (20)** | **22.66 (17)** | **41.64 (17)** | **27.54 (13)** | **16.15 (17)** | **8.01 (14)** |
| | DMI† | 45.64 (41) | 25.20 (35) | 43.40 (31) | 32.42 (27) | 38.81 (N/A) | 23.83 (N/A) |
| | TMI† | 45.04 (39) | 25.20 (38) | 45.43 (37) | 34.77 (30) | 41.13 (N/A) | 25.59 (N/A) |
| | VMI† | 50.55 (N/A) | 28.71 (44) | 50.76 (N/A) | 37.11 (44) | 21.97 (39) | 11.72 (32) |
| | SVRE† | 59.12 (N/A) | 32.81 (N/A) | 60.37 (N/A) | 42.77 (N/A) | 36.11 (N/A) | 19.53 (136) |
| $\ell_2 : 0.5$ | BPDA | 79.36 (N/A) | 45.31 (110) | 86.41 (N/A) | 58.40 (N/A) | 75.02 (N/A) | 46.68 (147) |
| | DA/AA | 79.92 (N/A) | 46.29 (121) | 85.80 (N/A) | 59.57 (N/A) | 74.96 (N/A) | 46.48 (135) |
| | PGD | 78.38 (N/A) | 44.53 (100) | 83.57 (N/A) | 56.25 (96) | 72.91 (N/A) | 43.95 (75) |
| | DH | **74.49 (46)** | **41.41 (44)** | **78.83 (46)** | **53.12 (46)** | **68.54 (38)** | **41.02 (39)** |

# 4 Experiments

## 4.1 Experimental Setup

**Baselines.** We evaluated the robustness of three diffusion-based purification defenses: DiffPure [22], GDMP [26], and LM [3]. For purification, we used a pre-trained score-based diffusion model [24] in DiffPure and GDMP and an EDM model [14] in LM. To ensure a fair comparison, we employed the WideResNet-70-16 [32] as the classifier across all tests.

We selected three state-of-the-art attack algorithms equipped with EOT [2] for baseline evaluation: BPDA [1], PGD [21], and AA [4]. For DiffPure and GDMP, AA was upgraded to DA [13] by incorporating deviated-reconstruction loss. Our Diffhammer adopts AA as the default attack algorithm. We conducted three restarts totaling 150 iterations to thoroughly evaluate the robustness of the model. Additional configurations for defense and attack are detailed in Appendix C.1. Substitute gradient attacks [22, 31] tailored to diffusion-based processes are found to be inferior to PGD with full gradients [16], so we leave comparisons with these methods in the Appendix C.4. Additionally, we evaluate verifiable DiffSmooth classifiers [34], and the results are presented in the Appendix C.3.

**Evaluation metrics**. Consistent with prior work, we use subsets of the CIFAR10 [15], CIFAR100 [15], and ImageNettete [12] (a subset of 10 easily classified classes from Imagenet [6], more suited for robustness evaluation) with sizes of 512, 512, and 256 as datasets, respectively. The evaluation protocol is $N$-evaluation with $N = 10$, with the average robustness (Avg.Rob, $1 - \sum_{i,j} \mathcal{A}_i^{(j)}/NS$) and worst-case robustness (Wor.Rob, $1 - \sum_i (\max_i \mathcal{A}_i^{(j)})/S$) as metrics. Here $\mathcal{A}_i^{(j)}$ indicates whether sample $j$ was attacked at the $i$-th evaluation, and $S$ is the dataset size. We also reported the iterations taken to reach 90% of the best attack effect among all attacks as a metric of efficiency. Experimental results for CIFAR100 can be found in Appendix C.2, and adversarial samples visualization are shown in Appendix C.9.

Table 2: Performance of attacks against diffusion-based purification on ImageNettete [12].

| | Defense | DiffPure [22] | | GDMP [26] | | LM [3] | |
|---|---|---|---|---|---|---|---|
| | Metrics | Avg.Rob (it.)↓ | Wor.Rob (it.)↓ | Avg.Rob (it.)↓ | Wor.Rob (it.)↓ | Avg.Rob (it.)↓ | Wor.Rob (it.)↓ |
| | Clean | 97.03 | 95.31 | 97.11 | 94.53 | 96.41 | 94.53 |
| $\ell_\infty : 4/255$ | BPDA | 58.98 (N/A) | 50.78 (N/A) | 57.66 (N/A) | 51.56 (N/A) | 28.28 (28) | 22.66 (21) |
| | DA/AA | 53.12 (N/A) | 46.09 (N/A) | 46.64 (N/A) | 39.06 (N/A) | 51.41 (N/A) | 42.19 (N/A) |
| | PGD | 54.30 (N/A) | 46.88 (N/A) | 48.28 (N/A) | 38.28 (N/A) | 55.31 (N/A) | 45.31 (N/A) |
| | DH | **38.36 (14)** | **31.25 (11)** | **33.98 (11)** | **28.91 (14)** | **26.25 (14)** | **21.88 (11)** |

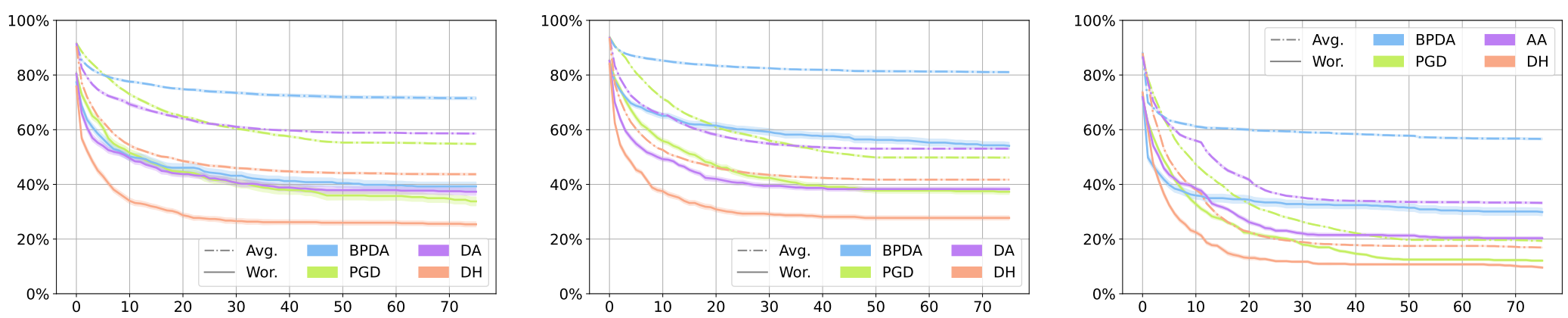

(a) DiffPure [22] (b) GDMP [26] (c) LM [3]

Figure 4: Avg.Rob and Wor.Rob for the first 75 steps of different attacks with $\ell_\infty : 8/255$.

### 4.2 Effectiveness and efficiency of DiffHammer

We examine the effectiveness and efficiency of various attack methods under different settings, with results for CIFAR and ImageNettete presented in Table 1,2, respectively. Our findings are as follows. (1) DiffHammer shows superior attack effectiveness across datasets and different norms ($\ell_\infty, \ell_2$) or attack budgets. This effectiveness arises from selective gradient aggregation, which avoids the gradient dilemma. Its performance improves significantly in high-stochasticity scenarios with pronounced gradient dilemmas, e.g., large-scale datasets like ImageNettete or GDMP with multiple purification rounds. (2) As shown in Figure 4 (other settings can be found in Appendix C.5), DiffHammer typically requires only 10-30 iterations to reach near-optimal results, allowing for rapid model robustness assessment. Other methods might occasionally bypass sampling from $\mathcal{S}_0$ for similar effectiveness, which incurs unnecessary computational costs. (4) Most models show robustness below 50% with 10 resubmits, indicating that a limited number of resubmits can undermine diffusion-based purification, raising concerns about their reliability in practical applications.

### 4.3 Gradient dilemma and transfer-based attack

We verify the gradient dilemma in diffusion-based purification by examining clustering effects and forgetting phenomena. Our EM algorithm clusters the gradients $\nabla_x \mathcal{L}_\phi$ into $\mathcal{S}_0$ and $\mathcal{S}_1$. The distribution of silhouette coefficients (SC) using cosine similarity, as displayed in Figure 5(a), indicates that gradients in these sets differ significantly. A direct result of this gradient dilemma is *attack forgetting*—where gradients $\nabla_x \mathcal{L}_\phi$ in consecutive iterations are inconsistent, causing the effects of previous attacks to be forgotten. As the ASR in $t-1$ iteration shown in Figure 5(b), DiffHammer maintains attack consistency by identifying $\mathcal{S}_1$, thus enhancing efficiency. We provide a toy example explaining the gradient dilemma in Appendix C.6. This dilemma may arise from a non-clustered data distribution (e.g., different breeds of dogs in the dataset), which imposes divergent gravitational pulls for purification and inconsistent perturbations needed to corrupt their features.

Transfer-based attacks offer a potential solution by treating adversarial samples like *models* and aiming to improve generalization on a *dataset* of $\phi$. As shown in 1, Data augmentation-based approaches, DMI [29] and TMI [8], provide improvement in some cases. VMI [27] and SVRE [30], which aim to reduce gradient variance, perform worse. This unexpected outcome is attributed to the gradient dilemma: generalizing to $\phi$ from $\mathcal{S}_1$ is beneficial, while generalizing to $\phi$ in $\mathcal{S}_0$ may be harmful. Thus, DiffHammer acts as a data-selection approach, contributing to data-centric design in transfer attacks.

Table 3: Effectiveness (Avg.Rob / Wor.Rob, %) of different attacks on robust model with purification under $\ell_\infty$ : $8/255$ settings, including TRADES [33] and AWP [28]. The original adversarial robustness (Rob) without purification is listed in parentheses.

| Classifier | AWP [28] (Avg.Rob / Wor.Rob, Rob:60.0) ↓ | | | TRADES [33] (Avg.Rob / Wor.Rob, Rob:53.1) ↓ | | |
|---|---|---|---|---|---|---|
| Purification | DiffPure [22] | GDMP [26] | LM [3] | DiffPure [22] | GDMP [26] | LM [3] |
| BPDA | 53.53 / **35.15** | 58.40 / 45.70 | 56.87 / 42.38 | 50.50 / **31.82** | 54.98 / **41.38** | 53.15 / **39.26** |
| PGD | 53.30 / 36.32 | 58.57 / 47.26 | 69.37 / 56.84 | 50.70 / 35.54 | 52.73 / 41.60 | 64.37 / 52.54 |
| DA/AA | **51.75** / 35.54 | 58.71 / 47.85 | 65.27 / 52.34 | **49.27** / 34.37 | 52.65 / 42.38 | 59.78 / 49.02 |
| DH | 53.65 / 35.55 | **58.26** / **45.68** | **56.48** / **40.43** | 49.55 / 34.37 | **51.57** / 41.40 | **53.10** / 39.64 |

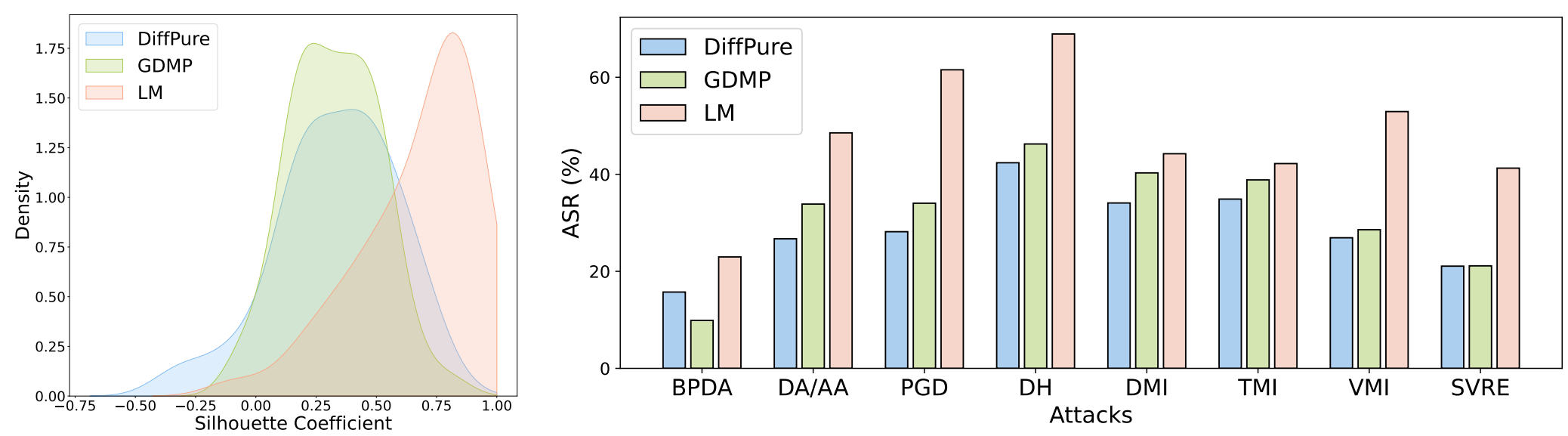

(a) Distribution of SC in $\mathcal{S}_0, \mathcal{S}_1$ (b) Attack successful rate (ASR) in previous $(t-1)$ iteration.

Figure 5: Clustering effects and forgetting phenomen in gradient dilemma ($\ell_\infty$ : $8/255$).

### 4.4 Robust classifier

Diffusion-based purification has been explored to enhance the robustness of adversarially trained (AT) models [22]. We revisited this concept using $N$-evaluation with TRADES [33] and AWP [28] as AT classifiers, as shown in Table 3. Our observations are as follows: (1) Robust models mitigate the gradient dilemma, resulting in similar performance for most attacks. This occurs because adversarial training implicitly regularizes the approximate gradient $\nabla_\phi \mathcal{L}_\phi$ [23], reducing the full gradient $\nabla_x \mathcal{L}_\phi$ of some $\phi$ towards zero. DiffHammer remains an effective evaluation tool, as diffusion-based purification occasionally reintroduces the gradient dilemma. (2) In most cases, diffusion-based purification weakens the robustness of the AT model. It improves robustness in a few instances but increases the risk of resubmit attacks by 11.7%-24.8%. Therefore, finding a better combination of diffusion-based purification and robust classifiers remains an open question.

### 4.5 Ablation study

Table 4: Performance difference (Avg.Rob / Wor.Rob, %) from default settings with $\ell_\infty$ : $8/255$.

| Ablations | DiffPure [22] | GDMP [26] | LM [3] |
|---|---|---|---|
| $\mathbb{E}q_\phi \nabla_\phi \mathcal{L}$ | +27.44 / +12.70 | +30.02 / +17.38 | +41.29 / +20.70 |
| $\mathbb{E}q_\phi \nabla_x \mathcal{L}$ | -3.52 / -1.17 | -3.09 / -1.35 | -0.63 / +0.32 |
| $\alpha = 0.2$ | +2.19 / +1.37 | +0.33 / -0.39 | +1.17 / +0.59 |
| $\alpha = 0.8$ | -0.16 / -0.39 | +0.63 / +1.17 | +1.23 / +1.17 |
| $N = 5$ | +1.44 / +2.34 | +1.52 / +2.73 | +7.48 / +4.89 |
| $N = 20$ | -1.41 / -1.77 | -0.47 / -0.59 | -0.51 / -0.47 |

We conducted ablation experiments on different components in the default settings (grafted gradient $\partial\hat{\phi}/\partial x \mathbb{E}(q_\phi \nabla_\phi \mathcal{L}_\phi)$, $\alpha = 0.5$, $N = 10$), with results shown in Table 4. The use of a selective approximate gradient $\mathbb{E}(q_\phi \nabla_\phi \mathcal{L}_\phi)$ leads to a significant performance drop, highlighting the importance of the purification gradient $\partial\hat{\phi}/\partial x$ in attacks. The grafted gradient causes only slight performance degradation compared to the full gradient $\mathbb{E}(q_\phi \nabla_x \mathcal{L}_\phi)$ but avoids a $\sim 10\times$ time burden. We quantify the time cost of different approaches in the Appendix C.7. A smaller hyperparameter $\alpha$ makes the algorithm rely more on the current iteration's gradient, and an appropriate $\alpha = 0.5$ achieves a better tradeoff in memorizing and learning. We tested effectiveness at 10-evaluations with $N$ samples of $\phi$, finding that a proper $N = 10$ efficiently identifies $\mathcal{S}_1$ for

evaluation. When $N = 20$, attack effectiveness is slightly boosted with more time overhead. The importance of $N$-evaluation for resubmit risk estimation is verified in the Appendix C.8.

## 5 Discussion and Insights

Diffusion models remain a promising solution to the adversarial samples due to their fine-grained modeling of data distributions. We advocate for enhancing the robustness of diffusion-based purification through standardized and powerful evaluation methodologies. Here are some insights into purification-based defense and attack:

(1) For deploying stochastic defenses, defenders should consider the potential number of resubmissions $M$ by attackers and are advised to assess resubmit risk with $N$-evaluation where $N \geq M$. Additionally, robustness overestimation due to the gradient dilemma can be avoided by using selective attacks.

(2) On the attack side, adversarial samples are sensitive to defenses with high stochasticity. Thus, modern data-centric designs may help to enhance adversarial transferability.

(3) On the defense side, the goal of stochastic defense is to achieve $P(\mathcal{S}_0) \to 1$, meaning sample $x$ cannot be attacked for some purifications and cannot be attacked by a same adversarial noise for others. Therefore, purification needs to be coordinated with adversarial training at a more granular level.

## 6 Related Work

**Evaluation for adversarial purification.** As test-time adaptive defenses, the iterative process and stochasticity of diffusion-based purification complicate robustness evaluation. Typically, evaluation protocols report only the ASR from a single evaluation. The risk of resubmit attack, referred to the *nag factor*, is considered in [20] but is not fully explored. Regarding evaluation methods, although there are gradient estimators *AdjAttack* and attacks *score-attack* based on the characteristics of the diffusion process, both were found to be inferior to attacks based on the exact gradient [16], such as PGD [21], AA [4], or DiffAttack [13] with reconstruction loss. These methods mitigate stochasticity through the EOT [2], which inevitably inherits its shortcomings.

**Transfer-based attacks.** Transfer-based attacks aim to generalize attacks from seen defenses to unseen defenses. Data augmentation and improved optimization are the main approaches to enhance transferability. Attackers can craft adversarial noise resistant to defenses through input transformations [29] and gradient smoothing [8]. In the presence of multiple defenses, momentum [7], variance reducing [27, 30] enhance generalizability. Most transfer-based attacks assume that most defenses share vulnerabilities, enabling generalization in defenses, but this assumption may be invalid in diffusion-based purification.

## 7 Conclusion

In this paper, we address the limitations of EOT-based attacks in diffusion-based purification, attributed to the gradient dilemma, by introducing an effective and efficient method called DiffHammer. First, we propose a selective attack strategy that targets vulnerable purifications without encountering the gradient dilemma, enhancing evaluation efficiency through gradient grafting. Second, we incorporate $N$-evaluation within the loop to quantify the risk of achieving at least one successful attack in practice. We demonstrate DiffHammer's superior performance through comprehensive experiments and anticipate it will offer valuable insights for future designs of robust diffusion-based purification methods.

## Acknowledgements

We would like to thank the anonymous reviewers for their valuable suggestions on theory and experiments. This work is supported by the Project of Hetao Shenzhen-HKUST Innovation Cooperation Zone HZQBKCZYB-2020083.

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

# A Broader Impact and Limitations

**Broader Impact.** The vulnerability of deep neural networks to adversarial samples limits their broader application. Diffusion-based purification has shown surprising robustness against EOT-based attacks, making it a promising defense. In this paper, we identify that EOT-based attacks often overestimate model robustness due to the gradient dilemma. Additionally, traditional 1-evaluations are insufficient for assessing resubmit risk in stochastic defenses. We propose an effective robustness evaluation framework, *DiffHammer*, which circumvents the gradient dilemma and incorporates $N$-evaluation. Our selective attack approach applies to stochastic defenses beyond diffusion-based purification, enhancing the understanding of stochasticity in robustness and inspiring improved adversarial defenses. Our $N$-evaluation method uncovers resubmit risks in diffusion-based purification, providing comprehensive insights into potential threats from stochastic defenses and informing precautionary measures. Furthermore, we enhance attack efficiency through gradient grafting, enabling rapid iteration of attacks and defenses. We believe that the insights and techniques introduced in our work will guide the development of more robust diffusion-based purification methods, ultimately enhancing the security of machine learning systems.

**Limitations.** In this work, we present a new attack algorithm called *DiffHammer* that is effective against diffusion-based purification defenses. While this has potential misuse in safety-critical domains like healthcare or finance, the core insights uncovered could also inform the development of more robust diffusion-based defenses going forward. Specifically, we introduce a selective attack strategy to overcome the challenge of the gradient dilemma. However, the underlying reasons behind this gradient dilemma are still not fully understood. Our experiments are limited to image classification, and while our approach can extend to other tasks, specific task-related designs need exploration. Overall, this research uncovers vulnerabilities in current diffusion-based defenses, but the findings could also spur innovations to make these defenses more secure in the future.

# B Proofs

## B.1 Proof of Thm. 1

**Theorem** (Thm. 1 in the main text)**.** *Suppose $\phi$ can be divided into two sets, $A$ and $B$. The loss functions w.r.t. $r$ in these sets are defined as $\mathcal{L}_A(r) := \sigma^{-1}(P_{\phi\in A}(\mathcal{A} \mid r))$ and $\mathcal{L}_B(r) := \sigma^{-1}(P_{\phi\in B}(\mathcal{A} \mid r))$, which are $m_A$ and $m_B$ strongly concave, respectively. If the distance between their optimal points $r_A$ and $r_B$ satisfies $\|r_A - r_B\|_2^2 \geq 8\max\{P(A)\mathcal{L}_A(r_A), P(B)\mathcal{L}_B(r_B)\}/m$ where $m := \min\{P(A)m_A, P(B)m_B\}$, the EOT-based attack is less effective than a simple attack targeting either $A$ or $B$.*

*Proof.* The obejective for EOT-based attacks is $\max_r P(A)\mathcal{L}_A(r) + P(B)\mathcal{L}_B(r)$, while the obejective for attacks targeting either $A$ or $B$ is $\max_r P(A)\mathcal{L}_A(r)$ or $\max_r P(B)\mathcal{L}_A(B)$ and the optimal points are $r_A$ and $r_B$, respectively. For any $r \in \mathbb{R}^d$, there exist $\theta \in \mathbb{R}$ s.t. $r = r_A + \theta(r_B - r_A)$. Therefore, the following inequality holds:

$$
\begin{aligned}
&P(A)\mathcal{L}_A(r) + P(B)\mathcal{L}_B(r) \\
&\leq P(A)\mathcal{L}_A(r_A) + P(B)\mathcal{L}_B(r_B) - \frac{P(A)m_A}{2}\|r - r_A\|_2^2 - \frac{P(B)m_B}{2}\|r - r_B\|_2^2 \text{ (strongly concave)} \\
&\leq 2\max\{P(A)\mathcal{L}_A(r_A), P(B)\mathcal{L}_B(r_B)\} - \frac{1}{2}m(\|r - r_A\|_2^2 + \|r - r_B\|_2^2) \\
&= 2\max\{P(A)\mathcal{L}_A(r_A), P(B)\mathcal{L}_B(r_B)\} - \frac{1}{2}m[\theta^2 + (1-\theta)^2]\|r_A - r_B\|_2^2 \; (r = r_A + \theta(r_B - r_A)) \\
&\leq 2\max\{P(A)\mathcal{L}_A(r_A), P(B)\mathcal{L}_B(r_B)\} - \frac{1}{8}m\|r_A - r_B\|_2^2 \quad (\left[\theta^2 + (1-\theta)^2\right] \geq \frac{1}{4}) \\
&\leq \max\{P(A)\mathcal{L}_A(r_A), P(B)\mathcal{L}_B(r_B)\}. \quad \text{(condition on } \|r_A - r_B\|_2^2\text{)}
\end{aligned}
\tag{12}
$$

□

### B.2 Proof of Thm. 2

**Theorem** (Thm. 2 in the main text)**.** *Let the sample's robustness in $M$ resubmit attacks $\mathcal{A}_1, \cdots, \mathcal{A}_M$ be denoted as $Rob := P(\mathcal{A}_1 = \cdots = \mathcal{A}_M = 0)$. There exists a uniformly minimum-variance unbiased estimator (UMVUE) for Rob if and only if the number of evaluation trials $N \geq M$. When $N \leq M$, the maximum likelihood estimator (MLE) tends to overestimate Rob in expectation, i.e., $\mathbb{E}(\widehat{Rob}^{(N)}_{MLE}) \geq Rob$.*

*Proof.* First, we provide detailed proof for the uniformly minimum-variance unbiased estimator (UMVUE) case, where we shall use the Lehmann–Scheffé theorem.

**Theorem 3** (Lehmann–Scheffé theorem [17])**.** *Let $X_1, \cdots, X_N$ be random variables from an unknown distribution $f(x;\theta)$ where $\theta \in \Omega$ is a parameter in the parameter space. Suppose $T(X)$ is a sufficient and complete statistic. If and only if condition for $\eta(T)$ to be UMVUE of $g(\theta)$ is that $\mathbb{E}_\theta[\eta(T)] = g(\theta)$.*

We denote the results of $N$ resubmissions as $\mathcal{A}_1, \cdots, \mathcal{A}_N$, and the probability of successful attack for each resubmission as $p$. Then $\mathcal{A}_1, \cdots, \mathcal{A}_N$ follows Bernoulli distribution, i.e. $\mathcal{A}_i \overset{\text{i.i.d.}}{\sim} Bin(1,p)$. According to the definition, $Rob = (1-p)^M$. Since the Bernoulli distribution is one of the exponential family of distributions, the sample sum $\sum_{i=1}^N \mathcal{A}_i$ is known to be the complete and sufficient statistics. Naturally, we have

$$\mathbb{E}\left[(1-\mathcal{A}_1)\cdots(1-\mathcal{A}_M)\right] = (1-p)^M, \tag{13}$$

If $N \geq M$, the conditional expectation can be written as

$$\begin{aligned}
&\mathbb{E}\left[(1-\mathcal{A}_1)\cdots(1-\mathcal{A}_M) \mid \mathcal{A}_1 + \cdots \mathcal{A}_N = n\right] \\
&= \mathbb{P}\big(\mathcal{A}_1 = \cdots = \mathcal{A}_M = 0 \mid \mathcal{A}_1 + \cdots \mathcal{A}_N = n\big) \\
&= \tbinom{N-M}{n} / \tbinom{N}{n},
\end{aligned} \tag{14}$$

which do not depend on $p$ since the sum $\mathcal{A}_1 + \cdots \mathcal{A}_N$ is sufficient, and therefore can be used as an estimator according to Rao-Blackwell Theorem:

$$\widehat{Rob}^{(N)}_{UMVUE} = \tbinom{N-M}{n} / \tbinom{N}{n}. \tag{15}$$

Due to the uniqueness of the UMVUE, the estimator is only available when $N \geq M$.

Next, we show that when $N \leq M$, the maximum likelihood estimator (MLE) tends to overestimate $Rob$ in expectation, i.e., $\mathbb{E}(\widehat{Rob}^{(N)}_{MLE}) > Rob$. Following the notation above, the MLE estimator for $p$ is $n/N$. Due to the transformation invariance of the MLE, the MLE estimator for $Rob := (1-p)^M$ is $(1-n/N)^M$; here, $n$ follows binomial distribution $Bin(N,p)$. Therefore, the expectation of the MLE estimator is

$$\begin{aligned}
\mathbb{E}(\widehat{Rob}^{(N)}_{MLE}) &= \mathbb{E}(1-\frac{n}{N})^M = \sum_{k=0}^{N} \tbinom{N}{k} p^k (1-p)^{N-k} (1-\frac{k}{N})^M \\
&= (1-p)^N + \sum_{k=1}^{N} \tbinom{N}{k} p^k (1-p)^{N-k} (1-\frac{k}{N})^M > (1-p)^N \geq (1-p)^M.
\end{aligned} \tag{16}$$

Therefore, estimating resubmit risk with MLE requires a larger $N$. When $N \leq M$, MLE will underestimate such risk. When $N$ goes larger, the MLE estimator is asymptotically normal. Since $\sqrt{N}(n/N - p) \xrightarrow{d} \mathcal{N}(0, p(1-p))$, the Delta method given function $g(p) = (1-p)^M$ implies $\sqrt{N}(g(n/N) - g(p)) \xrightarrow{d} \mathcal{N}(0, p(1-p)g'(p)^2)$, $\widehat{Rob}^{(N)}_{MLE}$ is asymptotically normal:

$$\sqrt{N}(\widehat{Rob}^{(N)}_{MLE} - Rob) \xrightarrow{d} \mathcal{N}(0, M^2 p (1-p)^{2M-1}). \tag{17}$$

□

## C Experiments

### C.1 Experimental Configuration

#### C.1.1 Defense configurations

**DiffPure [22].** DiffPure adds noise to potentially adversarial samples $x_0$ through the forward process:

$$x_{t^*} = \sqrt{\bar{\alpha}_{t^*}} x_0 + \sqrt{1 - \bar{\alpha}_{t^*}} \epsilon, \tag{18}$$

where $\alpha_t$ is a predefined scheduler and $\bar{\alpha}_t = \prod_{i=1}^{t} \alpha_i$, $\epsilon$ is a standard Gaussian noise from $\mathcal{N}(0, 1)$.

The reverse process gradually removes the Gaussian noise along with the adversarial noise:

$$x_{t-1} = \frac{1}{\sqrt{\alpha_t}} \left[ x_t - \frac{1 - \alpha_t}{1 - \bar{\alpha}_t} \epsilon_\theta(x_t, t) \right] + \sigma_t z \tag{19}$$

where $\epsilon_\theta(x_t, t)$ is a parameterized neural networks, $z$ is standard Gaussian noise in each step and standard deviation $\sigma_t = \frac{1-\bar{\alpha}_{t-1}}{1-\alpha_t}(1 - \bar{\alpha}_t)$.

In the experiment, $t^*$ is set to 0.1 ($\ell_\infty$) and 0.075 ($\ell_2$) in CIFAR [15], and in ImageNettete [12], $t^*$ is set to 0.05. We use the surrogate process proposed in [16], i.e., applying different time intervals in attack and defense to improve the efficiency. The time interval for the attack is set to 0.01 (CIFAR, $\ell_\infty$; ImageNettete), 0.015 (CIFAR, $\ell_2$). The time interval for defense is set to 0.002 (ImageNettete), 0.005 (CIFAR).

**GDMP [26].** Compared to DiffPure, GDMP adds gradient guidance for image similarity $\mathcal{D}(x_t, x_{t'})$ in forward and reverse processes to preserve semantic information. The update of $x_{t-1}$ can be written as:

$$x_{t-1} = \frac{1}{\sqrt{\alpha_t}} \left[ x_t - \frac{1 - \alpha_t}{1 - \bar{\alpha}_t} \epsilon_\theta(x_t, t) - s\sigma_t \nabla_{x_t} \mathcal{D}(x_t, x_{t'}) \right] + \sigma_t z \tag{20}$$

where $s$ is a scale of guidance and $x_{t'} = \sqrt{\bar{\alpha}_{t'}} x_0 + \sqrt{1 - \bar{\alpha}_{t'}} \epsilon$. In addition, GDMP observes that multiple rounds of purification with smaller $t^*$ help to improve robustness.

In the experiment, $t^*$ is set to 0.036 in CIFAR [15], and 0.03 in ImageNette [12]. The rounds of purification are 4 and 2, respectively. The time interval for the attack is set to 0.006 (CIFAR) and 0.003 (ImageNette). The time interval for defense is set to 0.003 (CIFAR), 0.001 (ImageNette).

**LM [3].** LM is the preprocessor in the robust diffusion classifier (RDC) that maximizes the lower bound of the log-likelihood through the following optimization process:

$$\min_{\hat{x}} \mathbb{E}_{\epsilon,t} \|\epsilon - \epsilon_\theta(\hat{x}_t, t)\|_2^2, \qquad s.t. \quad \|\hat{x} - x_0\|_\infty \leq \eta. \tag{21}$$

where $\hat{x}_t = \sqrt{\bar{\alpha}_{t'}} \hat{x} + \sqrt{1 - \bar{\alpha}_{t'}} \epsilon$ and $\eta$ is a predefined threshold.

The above optimization in the experiment was solved by the projected gradient method with a step size of 0.1 and 5 iterations. $t$ is uniformly sampled from a uniform distribution of 0.4-0.6. It is worth noting that the RDC consists of the LM and a diffusion classifier (DC). We find that the purifier LM is not sufficient to provide reliable robustness. We conjecture that the robustness of RDC mainly stems from the diffusion-based classifier.

#### C.1.2 Attack configurations

Each attack is performed with three restarts, consisting of 50 iterations per restart, totaling 150 iterations. Samples with an attack success rate below 50% are restarted to save time. We utilize the APGD [4] as the update algorithm for DiffHammer and BPDA [1]. The step size in PGD [21] is set to 0.007, and the momentum coefficient is set to 1. In VMI [27] and SVRE [30], we replace the gradient sampled from the neighborhood of $x$ for variance estimation with approximate gradients sampled on $\phi$, adapting to stochasticity defense. Other attacks employ default parameter settings.

Table 5: Performance of attacks against diffusion-based purification on CIFAR100.

| | Defense | DiffPure [22] | | GDMP [26] | | LM [3] | |
|---|---|---|---|---|---|---|---|
| | Metrics | Avg.Rob (it.)↓ | Wor.Rob (it.)↓ | Avg.Rob (it.)↓ | Wor.Rob (it.)↓ | Avg.Rob (it.)↓ | Wor.Rob (it.)↓ |
| | Clean | 53.18 | 26.52 | 64.20 | 40.23 | 68.09 | 46.29 |
| $\ell_\infty : 4/255$ | BPDA | 32.77 (37) | 6.45 (7) | 44.80 (N/A) | 14.65 (16) | 41.45 (12) | 19.34 (7) |
| | DA/AA | 30.57 (18) | 5.47 (5) | 41.09 (26) | 14.84 (11) | 47.58 (N/A) | 20.90 (20) |
| | PGD | 30.49 (16) | 5.66 (5) | 40.94 (24) | 14.65 (11) | 48.57 (N/A) | 21.09 (20) |
| | DH | **27.83 (10)** | **4.69 (4)** | **37.64 (9)** | **12.50 (4)** | **41.11 (11)** | **18.75 (5)** |

Table 6: Performance of attacks against DiffSmooth [34] on CIFAR10.

| Defense | $\ell_2 : 0.5$ | | $\ell_2 : 1.0$ | |
|---|---|---|---|---|
| Metrics | Avg.Rob (it.)↓ | Wor.Rob (it.)↓ | Avg.Rob (it.)↓ | Wor.Rob (it.)↓ |
| Clean | 84.63 | 74.02 | 70.31 | 42.19 |
| BPDA | 74.40 (N/A) | 53.13 (N/A) | 49.84 (96) | 12.50 (22) |
| DA | 74.22 (N/A) | 53.91 (N/A) | 50.59 (123) | 13.67 (38) |
| PGD | 66.04 (89) | 46.29 (82) | 47.05 (39) | 10.74 (29) |
| DH | **63.18 (42)** | **42.58 (46)** | **45.29 (31)** | **10.16 (21)** |

### C.2 Experimental results on CIFAR100

We evaluate the effectiveness of various attacks on diffusion-based purification using CIFAR-100 [15]. Unlike its predecessor, CIFAR10, which had 10 categories, CIFAR100 contains 100 categories, providing a greater fine-grained classification challenge. For consistency, we employ the EDM unconditional diffusion model [14] as the purifier and a pre-trained PyramidNet [10] as the classifier. Given the increased complexity, we set the attack budget to 4/255. As shown in Table 5, DiffHammer consistently achieves superior attack results and efficiency across all settings, indicating the persistence of the gradient dilemma in finer-grained purification.

Notably, finer classifications seem benefit less from diffusion-based purification, with classifiers with diffusion-based purification even demonstrating less than 50% robustness without adversarial attacks. This may be due to current diffusion models' limitations in fine-grained generation, potentially introducing vulnerabilities during reconstruction. Thus, designing diffusion models with enhanced alignment for improved robustness in fine-grained classification remains an open challenge.

### C.3 Experimental results on DiffSmooth

DiffSmooth [34] enhances certified and empirical robustness through a purified classifier in the inner loop and a sampling-based certification process in the outer loop. Since certified robustness provides a theoretical lower bound on model robustness, we focus on attacking the inner-loop purified classifier. DiffSmooth's purification mechanisms involve single-step purification and majority voting, which can mitigate the risk of resubmission. Unlike DiffPure, DiffSmooth uses a single update:

$$\begin{aligned} x_{t^*} &= \sqrt{\bar{\alpha}_{t^*}} x_0 + \sqrt{1-\bar{\alpha}_{t^*}} \epsilon \\ \hat{x}_0 &= \frac{1}{\sqrt{\bar{\alpha}_{t^*}}} (x_{t^*} - \sqrt{1-\bar{\alpha}_{t^*}} \epsilon_\theta(x_{t^*}, t^*)), \end{aligned} \tag{22}$$

where $t^*$ satisfies $\bar{\alpha}_{t^*}\sigma^2 = (1-\bar{\alpha}_{t^*})$ and $\sigma$ is a predifined noise scale. DiffSmooth applies local smoothing through Gaussian sampling in the neighborhood of purified $\hat{x}_0$, and results are determined by majority voting with scale $\sigma'$:

$$y = \arg\max \sum_{i=1}^{m} f(\hat{x}_0 + \delta_i), \qquad \delta_i \sim \mathcal{N}(0, \sigma'^2 I). \tag{23}$$

Table 7: Performance of substitute gradient based attacks against diffusion-based purification on CIFAR10.

| | Defense | DiffPure [22] | | GDMP [26] | | LM [3] | |
|---|---|---|---|---|---|---|---|
| | Metrics | Avg.Rob↓ | Wor.Rob ↓ | Avg.Rob↓ | Wor.Rob↓ | Avg.Rob↓ | Wor.Rob↓ |
| $\ell_\infty : 8/255$ | Score [31] | 77.70 | 43.36 | 82.46 | 56.84 | 64.04 | 39.06 |
| | Full [31] | 64.24 | 37.11 | 62.46 | 44.92 | 46.88 | 26.95 |
| | Adjoint [22] | 58.61 | 32.03 | 57.66 | 40.43 | N/A | N/A |
| | DH | **42.54** | **22.66** | **41.64** | **27.54** | **16.15** | **8.01** |

Random smoothing models are primarily used to defend against adversarial noise under the $\ell_2$ norm. We compared the impact of different attacks on DiffSmooth with settings of $\ell_2 : 0.5$ and $\ell_2 : 1.0$. For the $\ell_2 : 0.5$ setting, we used $\sigma = 0.5, \sigma' = 0.25$, applying ResNet-110 trained with Gaussian smoothing as the classifier. For the 1.0 setting, $\sigma = 0.25, \sigma' = 0.12$, with SmoothAdv-trained ResNet-110 as the classifier. A pretrained score-based diffusion model [24] acted as the purifier. As the result shown in Table 6, majority voting was found to suppress the gradient dilemma, reducing DiffHammer's impact, though it still achieved better attack results with significant efficiency gains.

While majority voting defends against occasional resubmissions, it also poses challenges: (1) It can make the stochastic defense model resemble a deterministic one, potentially reducing stochasticity benefits. (2) It imposes a significant time burden, intensified by the diffusion process's duration. DiffSmooth mitigates this with single-step purification, but this results in more fragile purification.

### C.4 Comparison with substitute gradients

Stochastic and iterative algorithms for diffusion-based purification yield challenging gradients computation, so a line of works aimed at approximating gradients in attack algorithms. This includes the Adjoint method [22] and Joint methods (score / full) [31].

Adjoint method obtain the gradient $\nabla_x \mathcal{L} = \sqrt{\bar{\alpha}_{t^*}} \nabla_{x_{t^*}} \mathcal{L}$ through an augmented SDE of Equation 5, which is solved as:

$$\begin{pmatrix} x_{t^*} \\ \nabla_{x_{t^*}} \mathcal{L} \end{pmatrix} = \texttt{sdeint}\left( \begin{pmatrix} \phi(x) \\ \nabla_\phi \mathcal{L} \end{pmatrix}, \tilde{\mathbf{f}}, \tilde{\mathbf{g}}, \tilde{w}, 0, t^* \right) \tag{24}$$

where `sdeint` is an SDE solver that sequentially takes six inputs: initial value, drift coefficient, diffusion coefficient, Wiener process, initial time, end time; and

$$\tilde{\mathbf{f}}([x; z], t) = \begin{pmatrix} \mathbf{f}(x, t) \\ \frac{\partial \mathbf{f}(x,t)}{\partial x} z \end{pmatrix}$$

$$\tilde{\mathbf{g}}(t) = \begin{pmatrix} -\mathbf{g}(t) \mathbf{1}_d \\ \mathbf{0}_d \end{pmatrix}$$

$$\tilde{w}(t) = \begin{pmatrix} -w(1-t) \\ -w(1-t) \end{pmatrix}$$

with $\mathbf{1}_d$ and $\mathbf{0}_d$ representing the $d$-dimensional vectors of all ones and all zeros, respectively.

The gradient in the joint attack (score) is a weighted sum of the approximate gradient and the estimated score function:

$$\text{sign}(\nabla_x \mathcal{L}) \simeq \lambda \text{sign}[s_\theta(x)] + (1 - \lambda) \text{sign}(\nabla_\phi \mathcal{L}), \tag{25}$$

where $s_\theta(x)$ is the estimated score function $\nabla_x \log p(x)$, $\nabla_\phi \mathcal{L}$ is the approximate gradient, and $\lambda$ is a balance factor set as 0.5 in the evaluation. The gradient in the joint attack (full) utilizes the difference between the original sample and the purified sample:

$$\text{sign}(\nabla_x \mathcal{L}) \simeq \lambda \text{sign}[\phi(x) - x] + (1 - \lambda) \text{sign}(\nabla_\phi \mathcal{L}). \tag{26}$$

While substitute gradient methods can produce gradients based on diffusion purification with lower time complexity, they have been found to inadequately evaluate robustness [16, 13]. We assess the

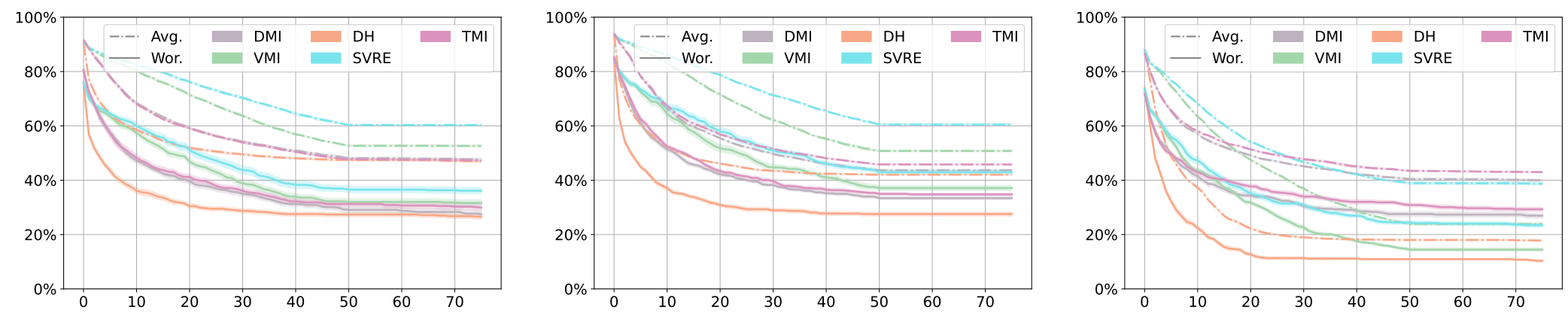

(a) DiffPure [22] (b) GDMP [26] (c) LM [3]

Figure 6: Avg.Rob and Wor.Rob for the first 75 steps of transfer based attacks in CIFAR10 with $\ell_\infty : 8/255$.

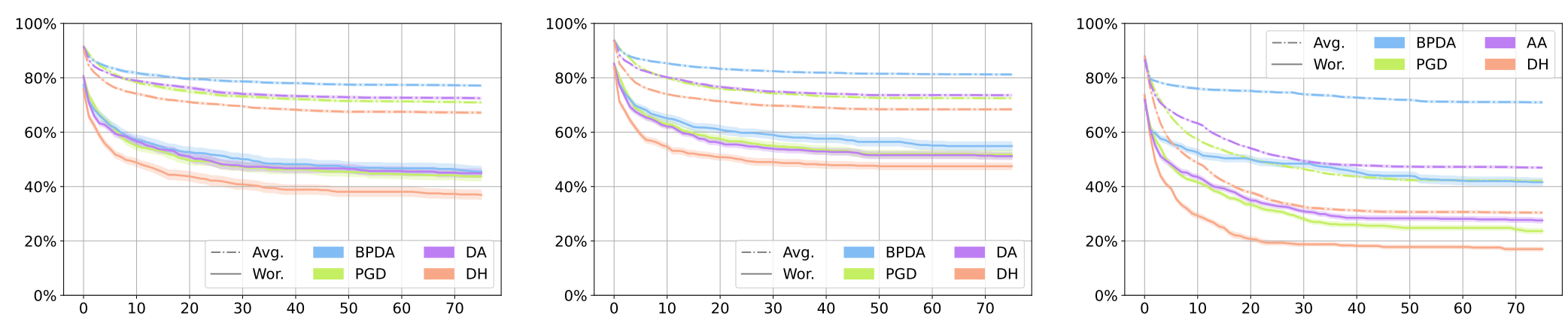

(a) DiffPure [22] (b) GDMP [26] (c) LM [3]

Figure 7: Avg.Rob and Wor.Rob for the first 75 steps in CIFAR10 with $\ell_\infty : 4/255$.

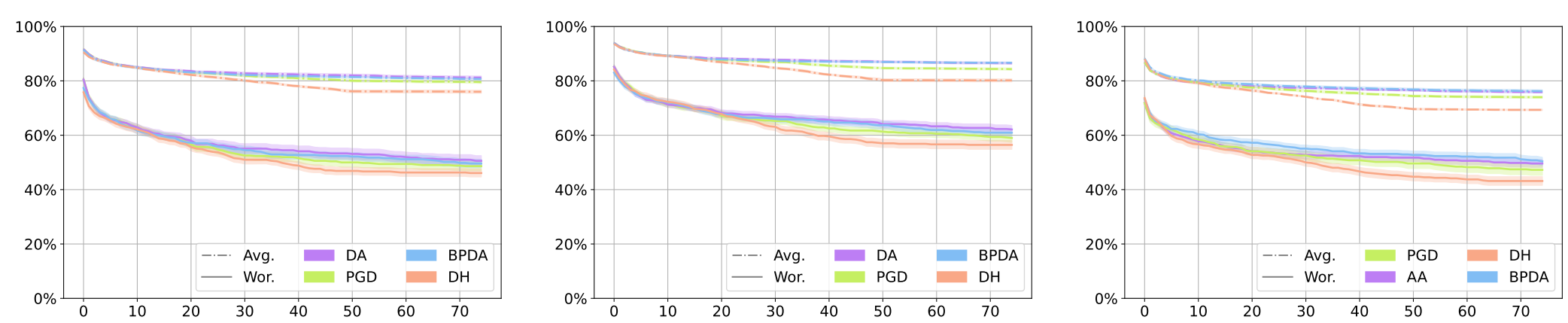

(a) DiffPure [22] (b) GDMP [26] (c) LM [3]

Figure 8: Avg.Rob and Wor.Rob for the first 75 steps in CIFAR10 with $\ell_2 : 0.5$.

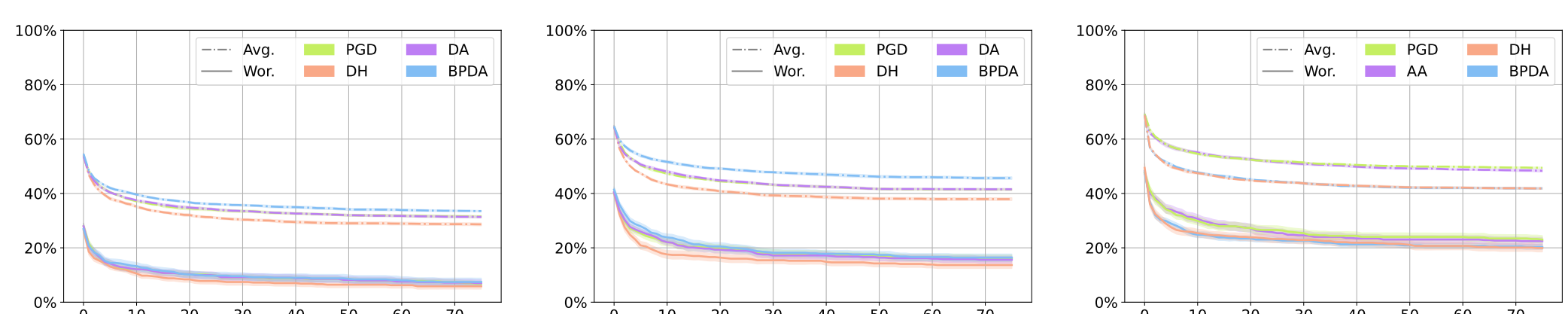

(a) DiffPure [22] (b) GDMP [26] (c) LM [3]

Figure 9: Avg.Rob and Wor.Rob for the first 75 steps in CIFAR100 with $\ell_\infty : 4/255$.

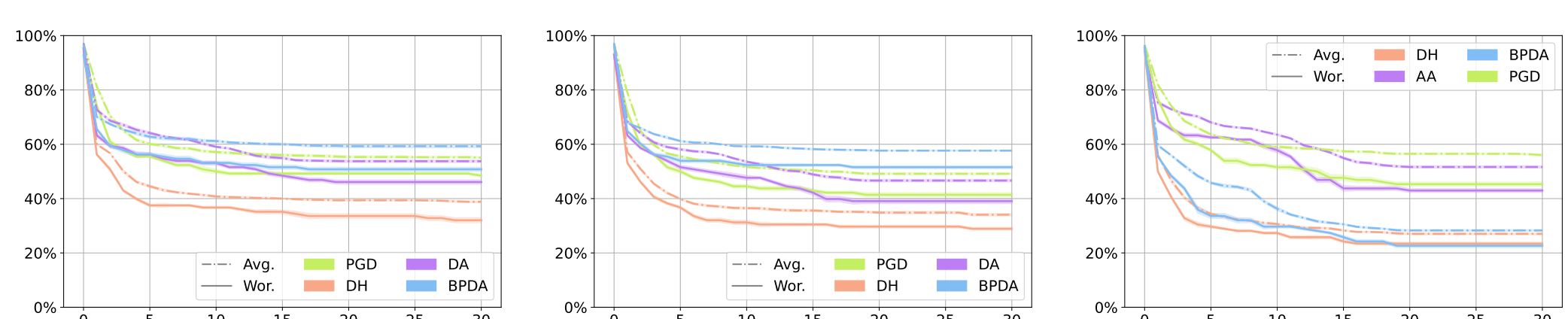

(a) DiffPure [22] (b) GDMP [26] (c) LM [3]

Figure 10: Avg.Rob and Wor.Rob for the first 75 steps in ImageNette with $\ell_\infty : 4/255$.

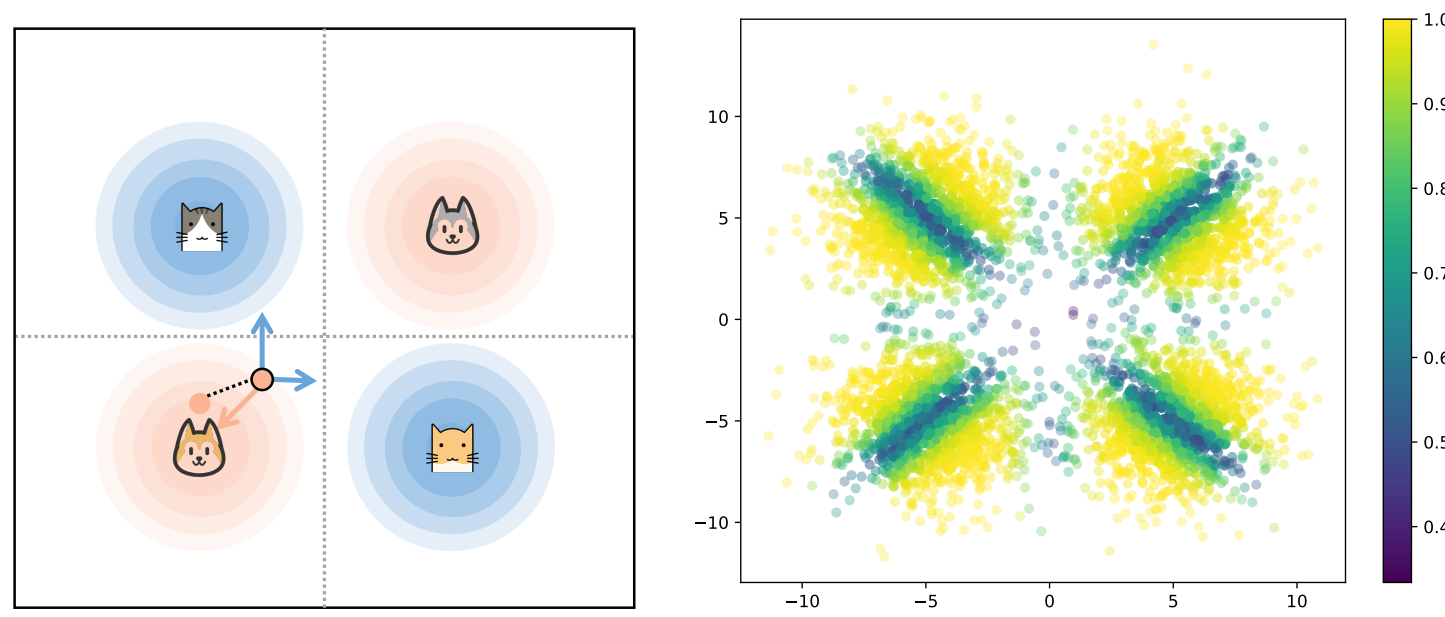

(a) Illustration of a toy example (b) Proportion of $\mathcal{S}_1$

Figure 11: A toy model of Gaussian mixing. Diffusion-based purification can be attacked toward unclustered features, leading to inconsistent gradients.

effectiveness of substitute gradients across different defenses, as shown in the Table 7. The substitute gradient fails to achieve 90% of DiffHammer effect in all settings, indicating a notable overestimation of model robustness. Specifically, the joint attack (score/full) only enhances the approximate gradient by the difference between samples and score function, lacking information from the diffusion process. Although the adjoint method theoretically provides an accurate gradient, in practice, numerical errors accumulate, leading to inaccurate estimates.

To address this, we utilize computational graph reconstruction proposed in [13] to obtain accurate gradients for DiffPure [22] and GDMP [26]. We progressively backpropagate the gradient by saving intermediate variables during iterations. Additionally, we derive the gradient of the optimization process in LM [3] using the Hessian vector product trick. This approach enhances existing gradient-based attacks with accurate gradient computation, establishing a more reliable baseline.

### C.5 Attack process with different settings

We present performance curves of the attack under various settings. Figure 6 compares DiffHammer with a transfer-based attack, showing that DiffHammer achieves superior results and faster convergence. Momentum-based attacks like DMI [29] and TMI [8] demonstrate good efficiency. However, transfer-based attacks yield sub-optimal results due to attempts to generalize across all purifications. Figure 7 and 8 illustrate performance curves under different constraints ($\ell_\infty : 4/255$ and $\ell_2 : 0.5$). Notably, DiffHammer significantly improves performance and efficiency under the $\ell_\infty : 4/255$ setting. With a limited attack budget, DiffHammer leverages the EM algorithm's fast convergence to quickly identify vulnerable set of $\phi$. In contrast, attack performances are more similar under the $\ell_2 : 0.5$ setting, likely because the $\ell_2$ constraint allows greater attack intensity on certain pixels, targeting shared vulnerabilities. Figure 9 and 10 compare attack effectiveness across datasets, revealing that DiffHammer excels in the ImageNette task, where complex purification processes seem more prone to introducing gradient dilemmas.

### C.6 Toy example of the gradient dilemma

To investigate the origins of the gradient dilemma, we constructed a Gaussian mixture toy example where the two diagonal components belong to the same category as shown in Figure 11(a). We use ReFlow [19] as our diffusion model to produce near-linear purification trajectories. In this scenario, diffusion-based purification pulls samples slightly towards the origin (dashed line in Figure 11(a)) before diffusing them back into the data distribution (pink line in Figure 11(a)). Given that the optimal classifier is a heteroscedastic classifier aligned with the coordinate axes, a sample's gradient direction is either horizontal or vertical, depending on which cluster the noised sample is closer to.

We demonstrated the proportion of $\mathcal{S}_1$ for each sampled point $x$ in Figure 11(a), where values nearing 50% indicate the presence of a gradient dilemma. Simulation results revealed that samples along the diagonal experience severe gradient dilemmas, which is consistent with intuition. In this toy example, the gradient dilemma arises from a non-clustered data distribution, which imposes divergent gravitational pulls for purification. Consequently, we hypothesize that real-world data distributions

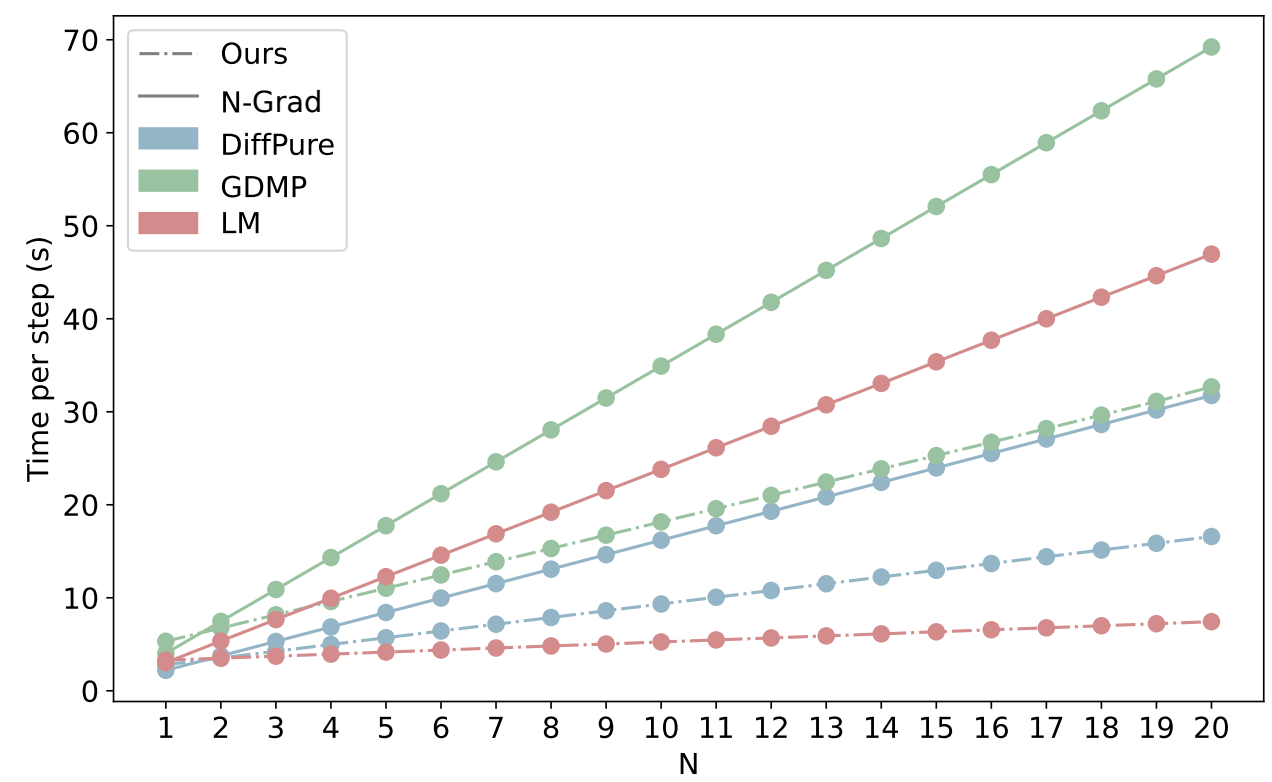

Figure 12: Time spent on gradient grafting and full gradient under different $N$.

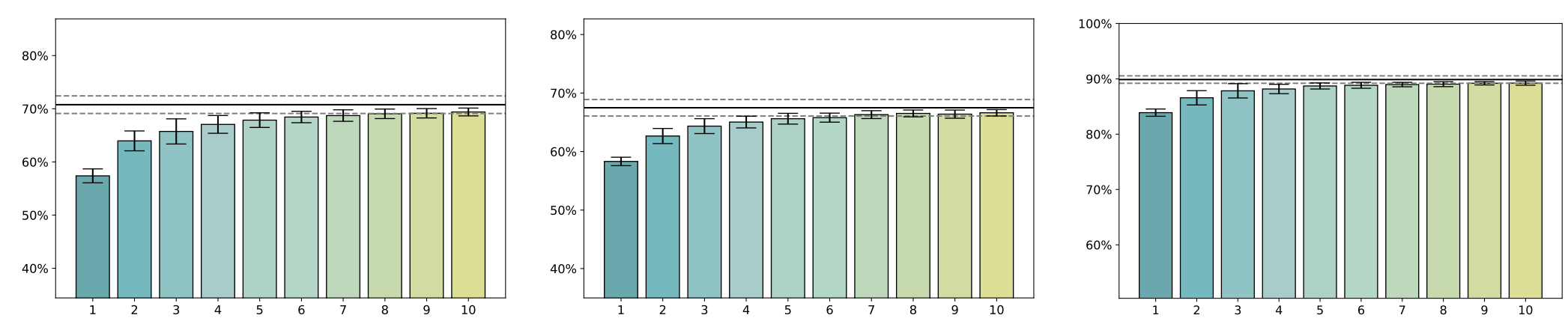

(a) DiffPure [22] (b) GDMP [26] (c) LM [3]

Figure 13: MLE estimation of the risk of 5 resubmit attacks for different $N$-evaluation. The reference value is represented by the solid line and its standard deviation by the dashed line.

may also exhibit non-clustered features. For instance, in the misclassification of a cat as a dog, considering the existence of various dog breeds, it is more efficient and effective to distort the features of one specific breed rather than all breeds collectively.

### C.7 Time complexity analysis

Our gradient grafting method enables the acquisition of gradients with lower time complexity for efficient attacks. We utilize only byproducts of $N$-evaluation, with the primary overhead being the computation of approximate gradients, which is less costly than full gradients. Our algorithm's computational cost involves $N$ approximate gradients and one full gradient, whereas a full gradient-based algorithm with 1-evaluation requires $N$ full gradients. When comparing computational times for varying $N$, we find that increasing $N$ slightly raises costs as the result shown in Figure 12. Compared to the $N$ full gradient computations in the EOT-based attack, DiffHammer computes the full gradient only once, and the cost of $N$ approximate gradients is acceptable. As $N$ increases, the efficiency gain from gradient grafting becomes more significant.

### C.8 Resubmit risk estimator

In a threat model where an attacker tries to obtain at least one successful attack by resubmitting, inappropriate evaluation can lead to an underestimated risk. For example, in labs that use facial recognition as authentication, users have up to five attempts. Therefore, the estimation of the 5-resubmit risk is critical for the defender. The defender can perform $N$ resubmit evaluations and use MLE as an estimator. We quantify the MLE-estimated risk for DiffHammer with $N$-evaluation, $N = 1, \dots, 10$ on a 5-resubmit attack, where the reference value is an empirical average of multiple trials. As the results in Figure 13 show, as the number of resubmissions increases, the estimation gets closer to the reference value with a smaller variance and tends to be stable and accurate when $N \geq M$. It is worth noting that when $N \ll M$, the estimation is either significantly underestimated ($N = 1$) or suffers from large variance ($N = 2$). Therefore, we recommend at least $N$-evaluation with $N \geq M$ as a means of evaluation.

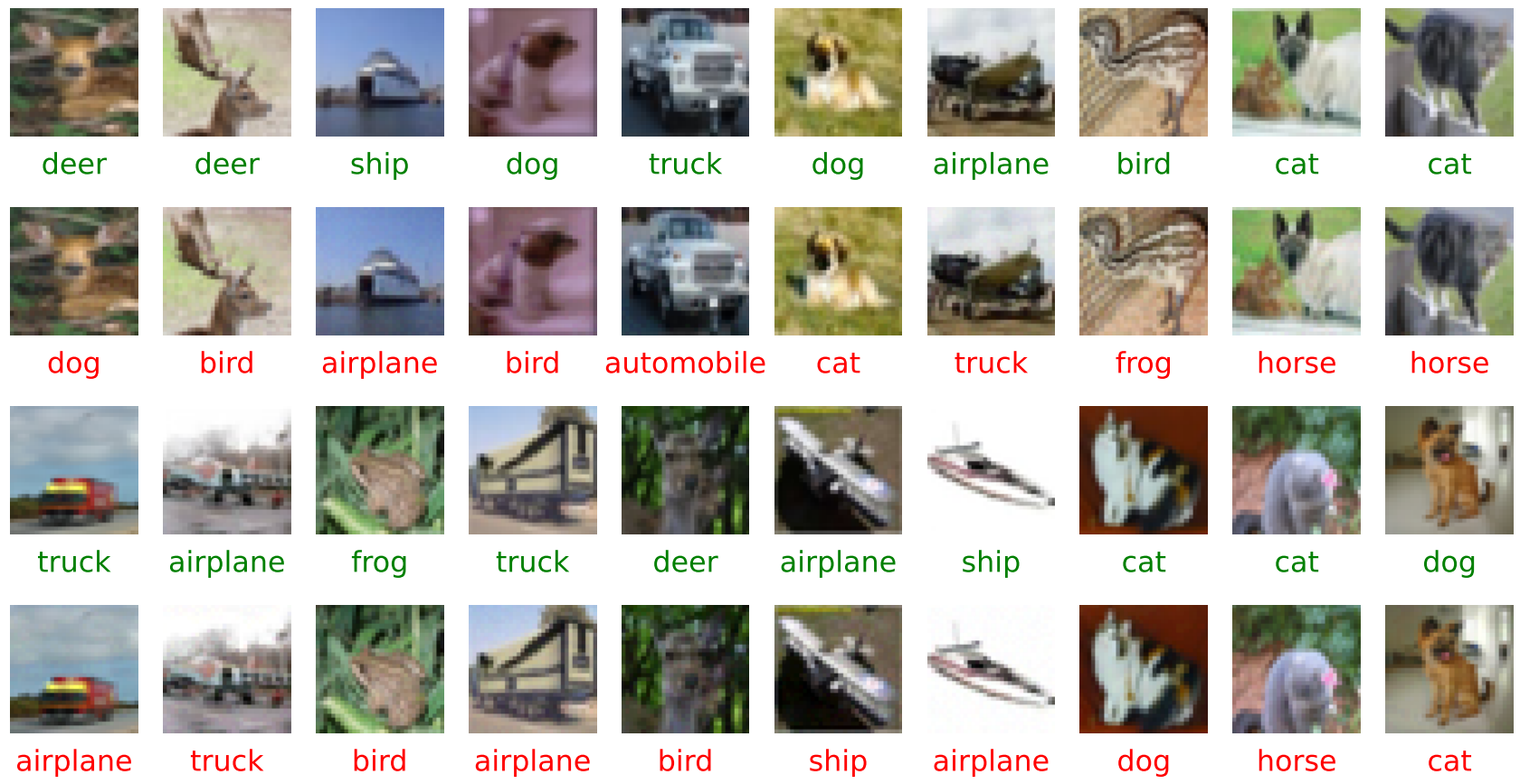

Figure 14: Example of visualization of adversarial samples on CIFAR10 ($\ell_\infty : 4/255$). Original labels are shown in green and adversarial labels are shown in red.

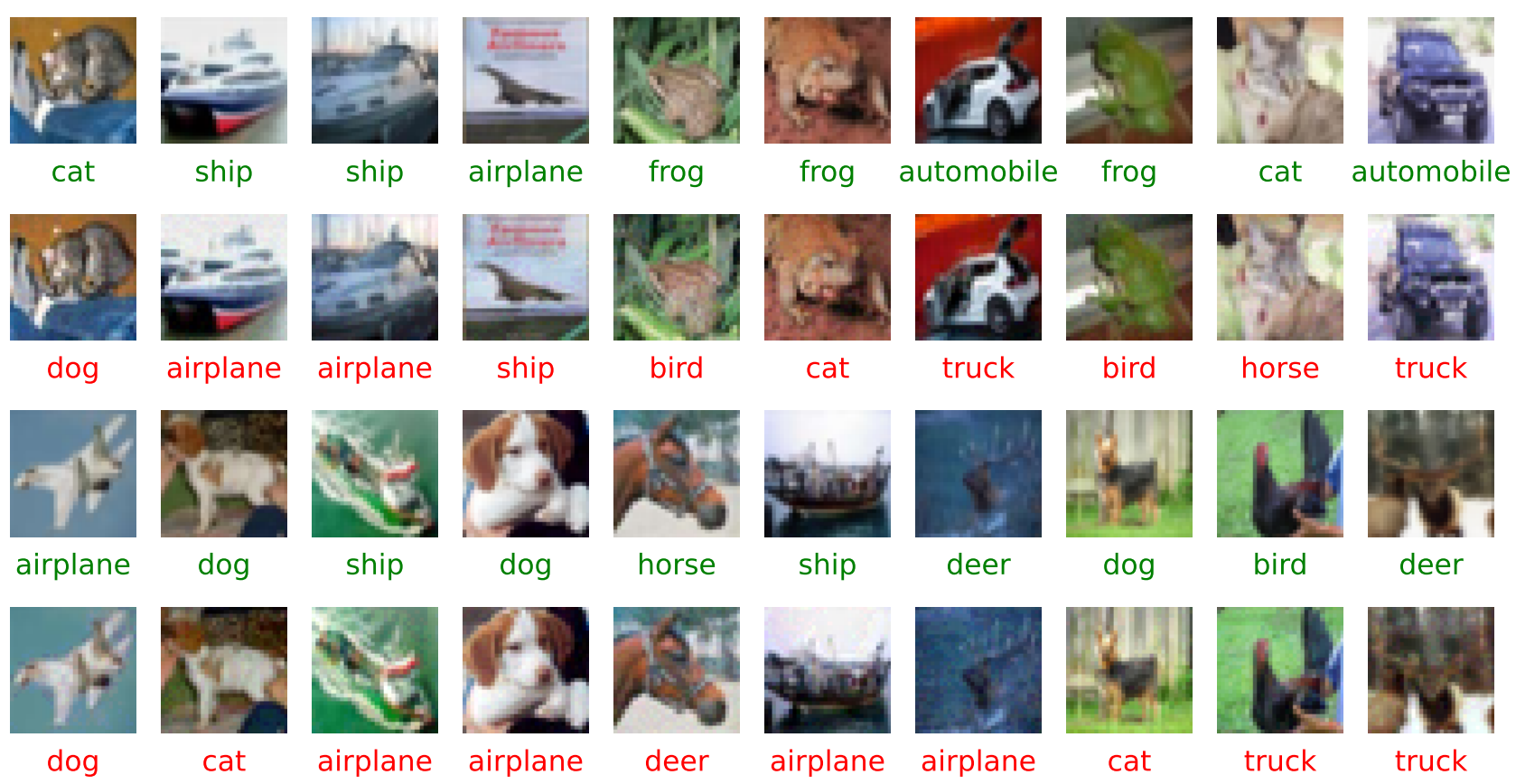

Figure 15: Example of visualization of adversarial samples on CIFAR10 ($\ell_\infty : 8/255$). Original labels are shown in green and adversarial labels are shown in red.

### C.9 Visualization

We visualize adversarial samples generated by DiffHammer against DiffPure [22] under various settings (CIFAR: Figure 14,15,16; CIFAR100: Figure 17; ImageNette: Figure 18). These samples are imperceptible even with a perturbation budget of 8/255. However, subtle adversarial perturbations can lead to significant differences in model decisions, highlighting the need for stronger adversarial defenses.

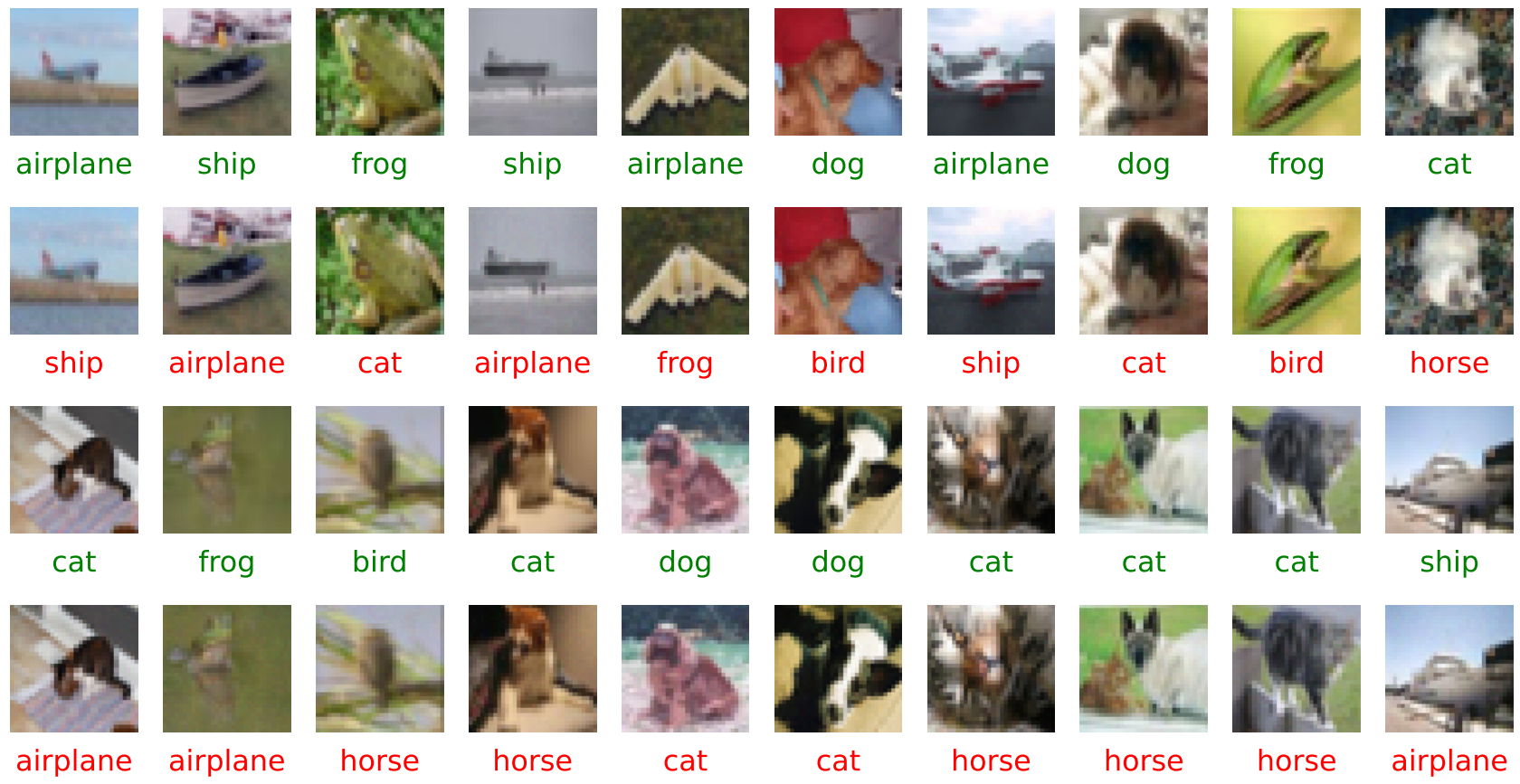

Figure 16: Example of visualization of adversarial samples on CIFAR10 ($\ell_2 : 0.5$). Original labels are shown in green and adversarial labels are shown in red.

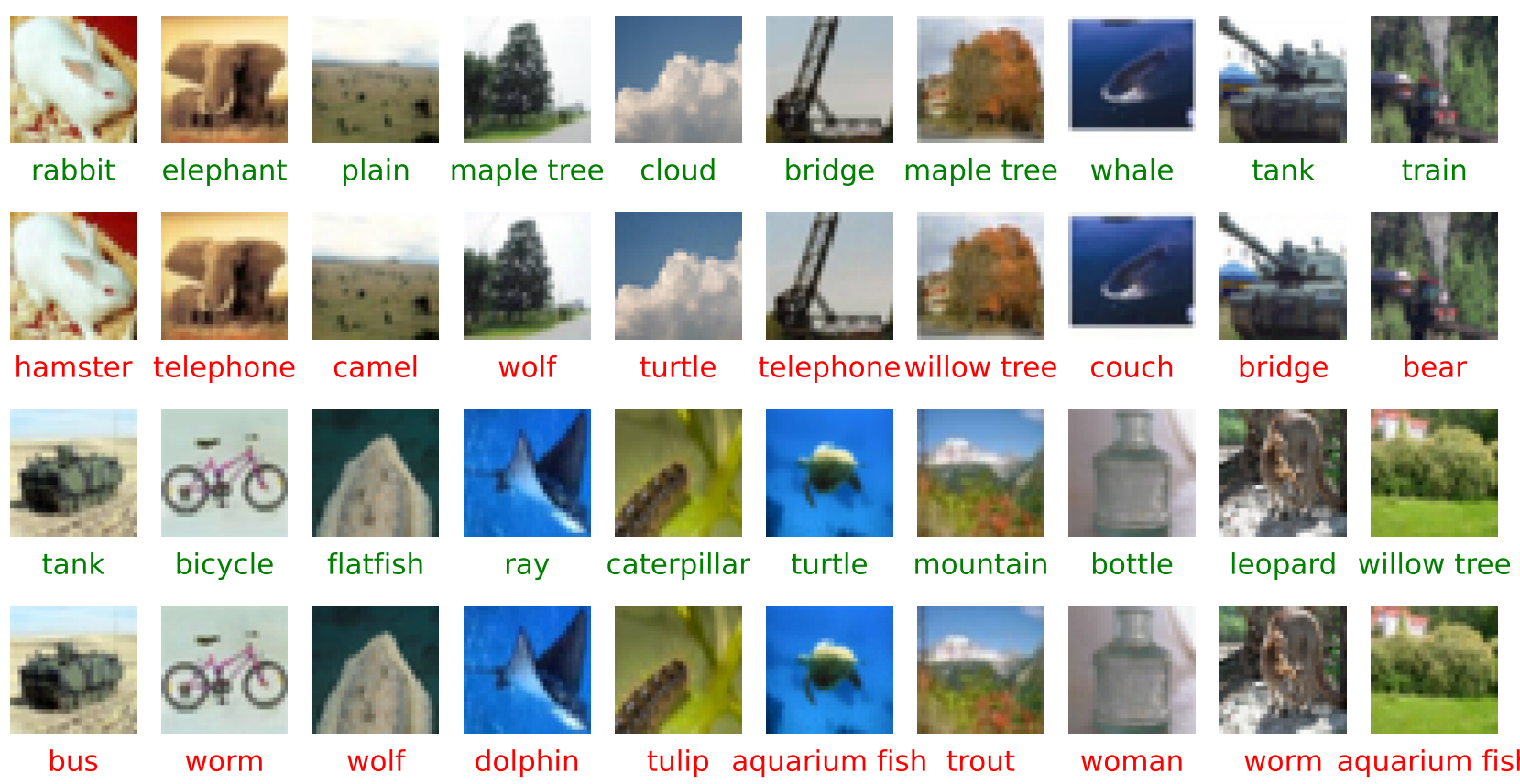

Figure 17: Example of visualization of adversarial samples on CIFAR100 ($\ell_\infty : 4/255$). Original labels are shown in green and adversarial labels are shown in red.

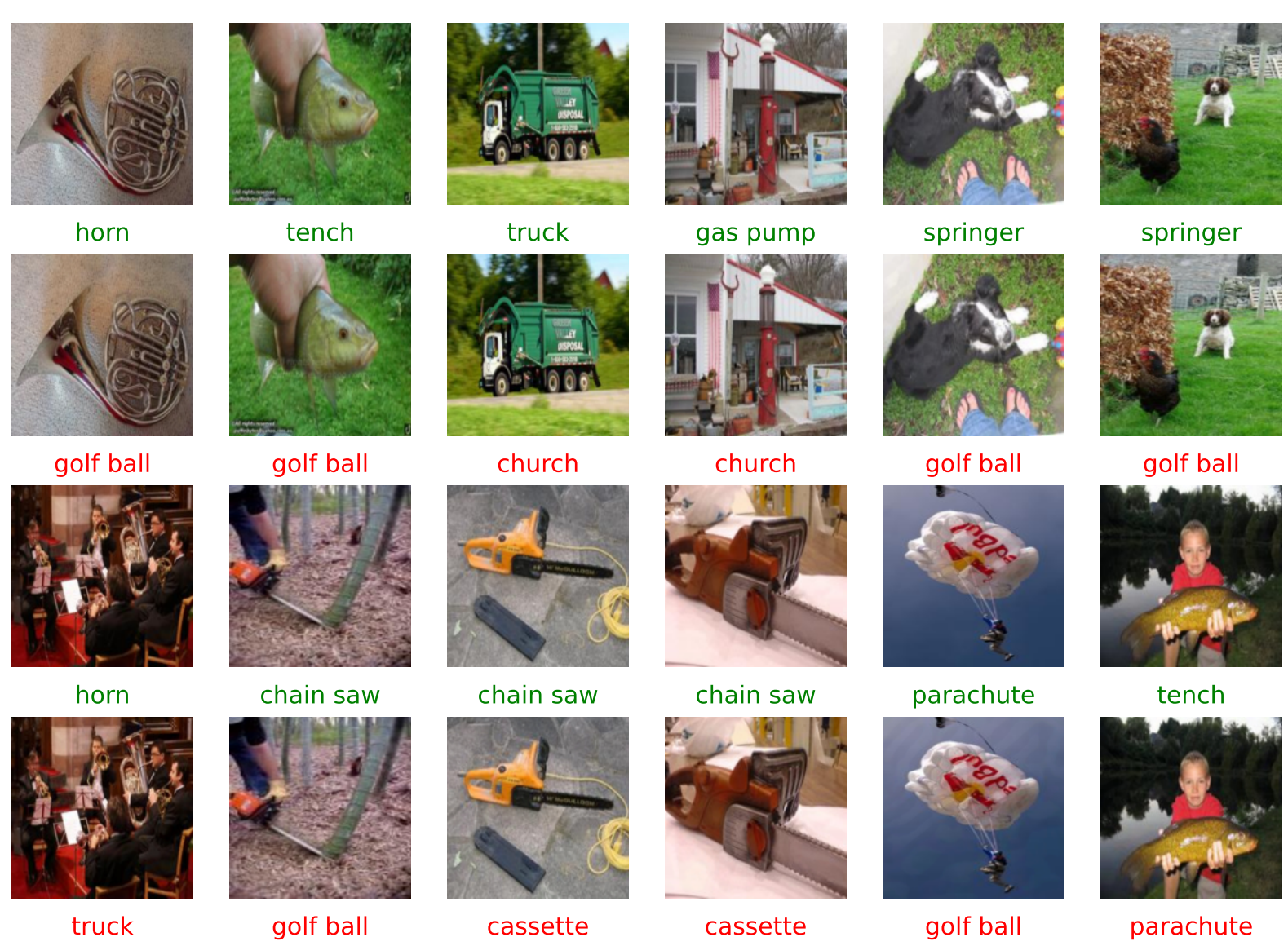

Figure 18: Example of visualization of adversarial samples on ImageNette ($\ell_\infty : 4/255$). Original labels are shown in green and adversarial labels are shown in red.

