# OpenReview forum: "DiffHammer: Rethinking the Robustness of Diffusion-Based Adversarial Purification"
_NeurIPS.cc/2024/Conference — NeurIPS 2024 poster_

### Official Review · Reviewer_ZG3o · 2024-07-12

**Soundness:** 2
**Presentation:** 3
**Contribution:** 2
**Rating:** 4
**Confidence:** 3

**Summary:**

This work proposes a new attack evaluation for diffusion-based purification methods: the 1 + N evaluation, which incorporates expectation maximization-based attacks and N-time evaluation. This method is helpful to evaluate the worst-case robustness of stochasticity-based defense methods.

**Strengths:**

1. This paper is well-written, with elaborative article organization and clear illustrations.
2. This paper gives a theoretical analysis of the advantages of N-time evaluation.

**Weaknesses:**

1.	One of the main conclusions on the advantages of N-evaluation is not that impressive. In fact, I think it is quite intuitive and the innovation is questionable.
2.	This setting will significantly increase the attack cost. If N is large enough, nearly all stochasticity-based defense methods have the risk of being broken. However, I do not think this is a practical and meaningful setting.
3.	The results cannot be compared with other methods directly. In addition to ASR, the absolute robustness should be reported in Table 1 so that these numbers can be directly compared with the results in other related literature [1, 2].
4.	Insufficient experimental results. Only the simple dataset, CIFAR-10, is considered. More datasets, especially ImageNet, should be incorporated.

[1] Robust Evaluation of Diffusion-Based Adversarial Purification, ICCV, 2023.

[2] Robust Classification via a Single Diffusion Model, ICML, 2024

**Questions:**

Please see comments above.

---

> ### Author Rebuttal · Authors · 2024-08-07
>
> Thank you for the valuable feedback, here's our response to the concerned problems.
> ## Advantages and costs of N-evaluation
>
> **Summary:** We were the first to propose using $N$-evaluation to enhance evaluation accuracy and improve attack effectiveness and efficiency. In terms of evaluation, we demonstrate that 1-evaluation underestimates resubmission risk. An appropriate $N$ (e.g., 10) is acceptable to attackers, making the risk realistically alarming. We'll include a discussion of this section in the paper.
>
> In terms of attack, we leverage the byproduct of $N$-evaluation, which incurs minimal additional overhead compared to the $N$-EOT attack. Our algorithm only require a small $N$ to overcome the gradient dilemma for sufficient attacks.
>
> Current robustness evaluations typically rely on 1-evaluation. However, in diffusion-based purification, the attack success rate for about 50% of samples ranges between 0 and 1, suggesting that attackers can undermine defenses by resubmitting. Evaluating resubmit attack risk depends on estimating attack probability, which 1-evaluation's non-zero-or-one result fails to provide. Even for samples with a 10-20% success rate (around 20% in our experiments), attackers can expect more than one successful attack out of 10 resubmissions with a feasible cost. Our analysis shows that 1-evaluation underestimate this risk statistically, as discussed in Section 3.1. Therefore, our proposed $N$-evaluation is practical amid the rise of diffusion-based defenses.
>
> Another contribution is using $N$-evaluation to aid the attack algorithm, which isn't burdensome because (1) increasing $N$ slightly raises costs, and (2) our algorithm remains stable without a large $N$. We use only the byproducts of $N$-evaluation, with the main overhead being the computation of approximate gradients, which is less costly than full gradients. Our algorithm's computational cost involves $N$ approximate gradients and one full gradient, whereas the EOT-based algorithm requires $N$ full gradients. Comparing computational times for varying $N$, we find that increasing $N$ doesn't significantly raise costs as shown in Rebuttal Figure.4. We also assess different $N$ values on algorithm performance, fixing evaluations at 10 for fairness. Results in Rebuttal Figure.5,6 show that a small $N$ (e.g. $N=5$) allows our algorithm to address the gradient dilemma and improve attack. Thus, $N$-evaluation effectively enhances attack efficiency at minimal cost.
> ## Direct comparison with other papers
> We computed robustness metrics using 1-ASR for direct comparison with other studies. Our differing results arise because the original paper reports the best historical robustness in 1-evaluation, while we focus on N-evaluations. Let $M$ denote iterations, and $n^{(N)}_m$ the successful attacks in the $m$-th iteration's N-evaluation. Our Avg.Rob is $1 - \max_m \sum n^{(N)}_m / N$, Wor.Rob is $1 - \max_m 1(n^{(N)}_m >0)$, while their Rob is $1 - \max_m 1(n^{(1)}_m >0)$. Their Rob is more akin to our Wor.Rob but from different samples, which can't assess model robustness for a given sample. After comparing methods consistently, most attack results align with the paper, except GDMP shows lower robustness due to the smaller WRN-28-10 classifier compared to DiffPure's WRN-70-16. To ensure fair benchmarking, we use WRN-70-16. Replacing it with WRN-28-10 reproduces GDMP results. Inconsistencies with LM results are due to numerical explosions in gradient computation which may disable the attack, we resolved it by gradient clipping.
>
> Table 1: Comparison with other papers.
>  eps |Defense|Attack|Avg.Rob|Wor.Rob|Rob|Rob (reported)
> -|-|-|-|-|-|-
> 4|	DiffPure|	DiffAttack|	70.80|	40.04|	65.04|	67.19
> 4	|DiffPure|	PGD|	71.27	|39.06|	65.82|	N/A
> 4	|DiffPure	|DiffHammer	|69.06|	35.94|	63.48|	N/A
> 4	|GDMP	|DiffAttack	|73.98	|52.54|	69.73	|N/A
> 4	|GDMP	|PGD	|77.77|	55.08	|74.61|	N/A
> 4	|GDMP	|DiffHammer	|72.29	|51.56|	68.75|	N/A
> 4|	LLHD_maximize|	AA|	49.22	|27.34|	44.14|	N/A
> 4|	LLHD_maximize|	PGD|	59.04|	33.40|	54.30|	N/A
> 4|	LLHD_maximize|	DiffHammer|	45.76|	20.31|	39.84|	N/A
> 8|	DiffPure|	DiffAttack|	56.00|	33.01|	54.56|	59.38
> 8|	DiffPure|	PGD|	60.27|	33.01|	54.69|	55.82
> 8|	DiffPure|	DiffHammer	|51.99|	27.93|	45.90|	N/A
> 8|	GDMP|	DiffAttack|	54.90|	39.65|	51.56|	N/A
> 8|	GDMP|	PGD|	65.37|	47.66|	60.55|	46.84
> 8|	GDMP|	DiffHammer|	50.37|	34.96|	45.90|	N/A
> 8|	LLHD_maximize|	AA	|36.43	|18.75|	31.84|	71.68
> 8|	LLHD_maximize	|PGD	|48.48|	26.76|	44.34|	N/A
> 8|	LLHD_maximize|	DiffHammer|	30.61|	14.06|	26.76|	N/A
>
> ## Other dataset
>
> CIFAR10 is the primary dataset for evaluating adversarial defenses, so we focus on it. We also include results on other datasets, like restricted ImageNet and CIFAR100, where restricted ImageNet's 9 superclasses present more challenges. We reduced the attack budget to 4/255. Results show our algorithm's ASR significantly surpasses others, indicating the gradient dilemma's presence across datasets.
>
> Table 2: Experiment result on restricted Imagenet
>
> | Method | DiffPure |      | GDMP |      | LM   |      |
> | ------ | -------- | ---- | ---- | ---- | ---- | ---- |
> | ASR    | Avg.     | Wor. | Avg. | Wor. | Avg. | Wor. |
> | BPDA   | 41.0     | 49.2 | 42.3 | 48.4 | 71.7 | 77.3 |
> | DA/AA  | 46.9     | 53.9 | 53.4 | 60.9 | 48.6 | 57.8 |
> | PGD    | 45.7     | 53.1 | 51.7 | 61.7 | 44.7 | 54.7 |
> | DH     | 61.6     | 68.8 | 66.0 | 71.1 | 73.8 | 78.1 |
>
> Table 3: Experiment result on CIFAR100
>
> | Method | DiffPure |      | GDMP |       | LM   |      |
> | ------ | -------- | ---- | ---- | ----- | ---- | ---- |
> | ASR    | Avg.     | Wor. | Avg. | Wor.  | Avg. | Wor. |
> | BPDA   | 67.2     | 93.6 | 55.2 | 85.4  | 58.6 | 80.7 |
> | DA     | AA       | 69.4 | 94.5 | 58.9  | 85.2 | 52.4 | 79.1 |
> | PGD    | 69.5     | 94.3 | 59.0 | 85.4  | 51.4 | 78.9 |
> | DH     | 72.2     | 95.3 | 62.4 | 87.50 | 54.4 | 80.5 |

---

> > ### Comment · Reviewer_ZG3o · 2024-08-13
> >
> > Thanks for the rebuttal. Part of my concerns have been addressed, except for W2 and W4.
> >
> > W2: I still do not think that N-evaluation is a practical and meaningful setting considering that nearly all stochasticity-based defense methods have the risk of being broken if N is large enough. In addition, if N=5 or N=10 would be recommended, why don't we use N=100 or even larger to evaluate? Conversely, when we discuss robustness, we often report the accuracy under attacks, which is the expectation of accuracy over the dataset. 1-evaluations naturally meet this perspective.
> >
> > W4: High-resolution and large-scale datasets, in particular, the ImageNet-1K, should be evaluated to enhance the results.
> >
> > Based on these concerns, I think this work to be a borderline case and will keep my rating.

---

> > > ### Author Response · Authors · 2024-08-13
> > > **Response for W2 and W4**
> > >
> > > Thank you for your question and feedback, and our response to your concerns is as follows:
> > >
> > > **W2:** In scenarios such as login, authentication, etc., the reward for an attacker to obtain at least one attack is high, while attempts are costly or limited. In such scenarios, the defender needs to consider worst robustness under a practical  $N$-evaluation setting. In such a scenario-dependent setting with practical $N$, stochastic defenses are not so inevitably attacked and should maintain a certain level of robustness. For example, the attacker needs to pay for API calls to attack, or may be blocked for a while when 5 incorrect authentications are submitted. In these cases, threats with $N=5$ need to be considered, while threats with $N=100$ are extreme/impossible. In short, the choice of $N$ is attack scenario-dependent, and $N$-evaluation can be adapted to these scenarios by setting a proper $N$.
> > >
> > > When considering robustness in the average sense, the stochastic defense yields different outputs for each submit, so its accuracy should be expected not only for the data, but also for the submit. Thus robustness corresponds to our Avg. Rob metric, which also requires $N$-evaluation to estimate.
> > >
> > > The importance of $N$-evaluation in evaluating stochastic defenses is also discussed in \[1\]. Our contributions on $N$-evaluation are (1) we statistically prove that 1-evaluation underestimates the risk of resubmits, and (2) attacks based on $N$-evaluation are more effective and efficient compared to those based on EOT. Overall, $N$-evaluation helps to assess the both average or worst case robustness of stochastic defenses in a wider range of scenarios, and can be used as an aid for better attacks even in strict scenarios where defenders only consider 1-evaluation threats.
> > >
> > > **W4:** We tested the effectiveness of our algorithm on the high resolution dataset restricted-ImageNet, which is ImageNet reorganized into 9 classes. Images are still 256*256 and from ImageNet1k, with the only difference being that fewer classes make the attack more challenging, which is widely used for evaluating model robustness \[2-4\]. Otherwise the 1000 classes in ImageNet1k make the attack easy and the robustness of the model is almost 0. As the results are shown in Table 2, our method DH performs better in different defenses. Our improvement is even more significant in denoising-based DiffPure and GDMP, which suggests that diffusion models on large-scale data are more likely to introduce gradient dilemmas to the defense. Thus, DiffHammer can be used as a better evaluation tool in different datasets.
> > >
> > > We appreciate your feedback and will clarify our contribution and add experimental results from other datasets in the revision. We sincerely appreciate it if you could let us know any additional questions or concerns. We are keen to provide a satisfactory response.
> > >
> > > \[1\] Lucas K, Jagielski M, Tramèr F, et al. Randomness in ml defenses helps persistent attackers and hinders evaluators[J]. arXiv preprint arXiv:2302.13464, 2023.
> > >
> > > \[2\] Chen H, Dong Y, Wang Z, et al. Robust classification via a single diffusion model[J]. arXiv preprint arXiv:2305.15241, 2023.
> > >
> > > \[3\] Tsipras D, Santurkar S, Engstrom L, et al. Robustness may be at odds with accuracy[J]. arXiv preprint arXiv:1805.12152, 2018.
> > >
> > > \[4\] Yang Y Y, Rashtchian C, Zhang H, et al. A closer look at accuracy vs. robustness[J]. Advances in neural information processing systems, 2020, 33: 8588-8601.

---

> > > ### Author Response · Authors · 2024-08-14
> > >
> > > Hi, We kindly look forward to receiving your feedback to give us the opportunity to further improve our paper and address your questions. Thank you!

---

### Official Review · Reviewer_knbF · 2024-07-12

**Soundness:** 2
**Presentation:** 2
**Contribution:** 3
**Rating:** 6
**Confidence:** 5

**Summary:**

The paper proposes a new adversarial attack framework against diffusion-based purification defenses. The paper first explains the advantage of using N-time evaluation as the metric for randomized defenses. Then, it proposes an E-M based adversarial attack, which empirically shows SOTA ASR.

**Strengths:**

1. Attacking diffusion-based purification defenses (SOTA defenses) is noteworthy to the community.
2. The attack framework is novel to me, and the empirical evidence is sufficient.

**Weaknesses:**

I think the randomness of diffusion purification is worthy of noticing and exploring in an EM framework with distribution-level analysis, which is a viable way from a high level. However, the major concern is that the paper does not convey the attack analysis in the EM framework very clearly. It might be a problem of presentation, but additional explanations are definitely needed for a fair judgment of the method.

1. Some definitions are not clear and rigorous. Line 141: missing rigorous definition of $g$. Line 143 says that $g(\phi)$ depends on a cluster, but the formulation does not show dependence on the cluster. Also, definition of “cluster” is missing.
2. Since sec 3.2 is not quite clearly written to me, I guess that you assume the adversarial perturbations for a given sample $x$ for a given defense $\phi_i$ follows a Gaussian with mean $r$ and covariance $\Sigma$. Is that correct? If so, the Gaussian assumption should be clearly emphasized with motivations of why you use Gaussian. Will assumptions of other distribution work? Is Gaussian important here? Empirical evidence would be beneficial.
3. Here’s another question: if you assume $g(\phi_i)$ denotes Gaussian perturbation regarding $\phi_i$ (according to equation 6, which include multiple $\phi_i$), then there should be $N$ Gaussians for $N$ defenses $\phi_1,..,\phi_N$? But there is just one $r$ and $\Sigma$ here.
4. Also, line 156 indicates that $q_i$ denote $N$ independent distribution on $z$, then should there be $N$ Bernoulli distribution param $\alpha$ here?
5. $q^{(t)}$ seems not defined in line 159.
6. Line 160: what does optimizing $r$ with higher $z$ mean? Should it be optimizing $r$ towards a high objective in eq. (6)?
7. In line 143, the paper indicates that $g(\phi)$ follows a normal distribution, which means that $g(\phi)$ is a random vector. But in equation (8), $g(\phi_i)$ is treated like a deterministic vector. Should $g()$ be inside the expectation?
8. In E-step, the constant $C$ looks quite important in Equation (9), how to select it? What is the rationale for approximation with a constant?
9. Logging in problem is one practical case for N-times evaluation, but is there any practical scenario for image misclassification?

**Questions:**

Please refer to the weakness part for concrete questions.

**Limitations:**

Discussed in Appendix A.

---

> ### Author Rebuttal · Authors · 2024-08-07
>
> Thank you for the valuable feedback, here's our response to the concerned problems. We will describe our EM framework more clearly and rigorously in response to concerned problems, and we will also update this part of the exposition in our paper.
>
>  Given a sample $x\in\mathbb{R}^d$ and its label $y\in [K]$, a stochastic purification $\phi:\mathbb{R}^d\to \mathbb{R}^d$ and a deterministic classifier $f:\mathbb{R}^d\to \mathbb{R}^K$ will classify it as $f[\phi(x)]$. Denotes the normalized gradient $g(\phi):=\nabla_x \mathcal{L}(f[\phi(x)];y) /\lVert \nabla_x \mathcal{L}(f[\phi(x)];y) \rVert_2$, we suspect that $g(\phi)$ may obey a multi-peaked distribution, where $g(\phi)$ can be divided into clusters that have high intraclass similarity and low interclass similarity.
>
> Our goal is to find the cluster center with the highest attack success rate. At each step of the attack, we can observe $N$ $g(\phi_i)$ and $\mathcal{A}_i, i=1,...,N$ (whether $f\circ \phi_i$ is attacked or not), We identify the parameters in the distribution of $g(\phi)$ by maximizing the likelihood function of $g(\phi_i)$ and $\mathcal{A}_i$.
>
> We make the following assumptions about the distribution of $g(\phi)$: (1) There exists a master cluster of $g(\phi)$ that obeys a Gaussian distribution. Let the hidden variable $z$ denote whether $g(\phi)$ comes from this cluster, then $p(g(\phi)|z=1) = \mathcal{N}(r,\Sigma)$ and the prior distribution of $z$ is the Bernoulli distribution with parameter $\alpha$. The proportion $\alpha$, mean $r$, and variance $\Sigma$ of this cluster are the parameters of the underlying model that we desired. (2) We use a null information prior for whole $g(\phi)$, i.e., we assume that $p(g(\phi))=c$ and $c$ will be eliminated in subsequent analyses. We make these assumptions because the normal distribution is flexible in modeling similarity between $g(\phi)$, and the zero-information prior avoids over-designing our model. Furthermore, these assumptions facilitate subsequent theoretical derivations, and we will show how certain steps of the algorithm can be modified to accommodate other assumptions.
>
> Since the hidden variable $z_i$ for each $g(\phi_i)$ is unknown, we use $q_i^{(t)}$ as an approximation of the $z_i$ posterior distribution at each stage $t$. Due to the properties of the Gaussian mixture model, the update involves only the mean of $q_i^{(t)}$ (i.e., the posterior mean of $z_i$: $\mathbb{E}[z_i|g(\phi_i)]=p(z_i=1|g(\phi_i))$), so in practice, $\mathbb{E}q_i^{(t)}$ can be viewed as an approximation of the posterior probability that $g(\phi_i)$ belongs to the main cluster. Then in M-step, $\mathbb{E}q_i^{(t)}$ is fixed and the parameters $r,\Sigma,\alpha$ are updated; in E-step, $\mathbb{E}q_i^{(t)}$ is updated to approximate $p(z_i=1|g(\phi_i))$.
>
> **M-step.** In M-step, maximizing our objective (Eq. 6 in the paper) can be achieved by gradient ascent in Eq. 8 in the paper, where the observed $g(\phi)$ is weighted by $\mathbb{E}q_i^{(t)}$, and $r$ moves towards the weighted center.
>
> **E-step.** As we discussed in Section 3.2.1 of the paper, we set $\Sigma$ as a hyperparameter, so we are only concerned with the update of $r$, where the previously assumed constant $c$ is eliminated.
>
> $$
> \mathbb{E}q_i^{(t)}=\frac{\alpha \mathcal{N}(g(\phi_i);r^{(t),\lambda^{-1}I})}{c}=A\exp(-\frac{\lVert g(\phi_i)-r^{(t)}\rVert_2^2}{2})=A\exp(\cos\langle g(\phi_i), r^{(t)} \rangle)
> $$
> where $A$ denotes (not a same) constants  independent of $i$. In the last term of Eq. 8 in the paper, due to the $\alpha$ of the denominator, it will become $\sum_i[\exp(\cos\langle g(\phi_i), r^{(t)} \rangle)g(\phi_i)]/\sum_i\exp(\cos\langle g(\phi_i), r^{(t)} \rangle) $. We can find that this is a result weighted by a normalized similarity function w.r.t. $g(\phi_i), r^{(t)}$, and the function $\exp\cos\langle \cdot, \cdot \rangle$ is the result of our assumption.
>
> Table1: DiffHammer in different assumption.
> | Attack | DiffPure |      |GDMP |      |LM||
> | ------ | ------- | ---- | ----- | ---- |-|-|
> | ASR    | Avg.    | Wor. | Avg.  | Wor. | Avg.  | Wor. |
> |DH2 | 53.5|73.4 | 56.0| 76.9 |78.0|87.9
> | DH| 53.8     | 75.2 | 57.9 | 72.5 | 83.3 | 91.4 |
>
> - Response to question 1: We will define $g(\phi)$ as the gradient and call a set with high intraclass similarity and low interclass similarity a cluster.
> - Response to question 2 & 8: We adopt the assumption of Gaussian distribution for flexibility and convenience, which provides a theoretical reference for our algorithm. We experimented with other customized similarity functions in the above E-step, e.g. we directly defined the similarity of $g(\phi_i)$ to $r$ as the expectation of the loss boost (the closer of these indicates the stronger the attack). As shown in Table 1 below, proper similarity function choices leads to good results in practice, which in theory corresponds to similar but different cluster modeling.
> - Response to question 3 & 4: Since $r,\Sigma$, and $\alpha$ are parameters of the underlying model, they are shared and unique among the $g(\phi)$.
> - Response to question 5: $q_i^{(t)}$ is an approximation of the $z_i$ posterior distribution at each stage $t$ and only $\mathbb{E}q_i^{(t)}$ is involved and updated.
> - Response to question 6: Maximizing the target with Eq. 6 is equivalent to making $r$ close to the $\mathbb{E}q_i^{(t)}$-weighted average $g(phi_i)$, which means that attacking a purifications with higher $\mathbb{E}q_i^{(t)}$.
> - Response to question 7: In Eq. 8, $g(\phi_i)$ denotes the observed random variable and is therefore deterministic.
> - Misclassification of models can constrain the deployment of models in safety-critical domains, for example, misclassification in autonomous driving can lead to catastrophic consequences. Diffusion-based sanitization serves as a promising solution, but may be undermined by resubmit attacks. Therefore, we reveal this risk to inspire subsequent research.

---

> > ### Comment · Reviewer_knbF · 2024-08-11
> > **Thanks for rebuttal**
> >
> > Thank the efforts of the authors in the rebuttal!
> >
> > I still want to make sure I correctly understand the assumptions here.
> >
> > "call a set with high intraclass similarity and low interclass similarity a cluster" is still not a rigorous and clear definition. Do you assume the gradients for successful attack follow a Guassian? And you just assume a single Gaussian since "$r,\Sigma$" are shared?

---

> ### Author Response · Authors · 2024-08-12
> **Response to the single Gaussian assumption**
>
> Thank you for your question. To define the cluster, we follow the definition in \[1\], where clusters are defined by "determine $m$ clusters (subsets) of individuals in $I$, and those individuals assigned to the same cluster are similar yet individuals from different clusters are different (not similar)".
>
> Our assumption is that, for a class of purifications that can be attacked by the same adversarial noise, the gradients of purifications in this class follow a Gaussian distribution with mean $r$. We estimate $r$ as an approximation to the adversarial noise. We set a null-informative prior on other purifications that are attacked (but not by $r$) or that are not attacked. Thus, we have only a shared set of Gaussian distribution parameters.
>
> In fact, it is not necessary to assume a Gaussian mixture model for all attacked  purifications and assign $m$ sets of parameters ($r_1,.... .r_m;\Sigma_1,... ,\Sigma_m;\alpha_1,... ,\alpha_m$) where $\alpha_1>\alpha_2... ,>\alpha_m$. The reason for this is that the EM algorithm is locally convergent and we are only concerned with the cluster mean $r_1$ that has the largest proportion ($\alpha_1$). Purification from $\mathcal{N}(r_1,\Sigma_1)$ is more likely to occur in the first few batches of observations due to the larger probability $\alpha_1$, which allows the estimated $r$ in EM algorithm to approach $r_1$ initially and optimize towards $r_1$ thereafter. Although the EM algorithm also has a probability of converging to a suboptimal, this also has a sufficient attack effect and avoids the additional overhead of maintaining $m$ sets of parameters. Therefore, we only assume a set of Gaussian parameters $r,\Sigma,\alpha$ to make our algorithm efficient and concise.
>
>
> \[1\] Duran B S, Odell P L. Cluster analysis: a survey[M]. Springer Science & Business Media, 2013.

---

> ### Comment · Reviewer_knbF · 2024-08-13
> **Thanks for response**
>
> Thanks for the authors' efforts for further clarification! My concerns are now addressed and I raise the score to 6. I hope the authors  can make the method part more clear and rigorous in revision.

---

> > ### Author Response · Authors · 2024-08-13
> >
> > We sincerely appreciate your acknowledgment and your positive feedback on our work! It is extremely valuable for helping us strengthen the paper.  We will make sure to clarify and enhance the rigor of the methods section in our revision.

---

### Official Review · Reviewer_wBtm · 2024-07-13

**Soundness:** 3
**Presentation:** 3
**Contribution:** 3
**Rating:** 5
**Confidence:** 3

**Summary:**

Diffusion-based purification methods have gained recognition for their robustness against adversarial attacks. However, concerns arise regarding the adequacy of current evaluation methods, particularly in addressing the gradient dilemma inherent in these techniques. This paper introduces DiffHammer, an advanced evaluation framework utilizing an EM-based attack and N-time evaluation to overcome these challenges. DiffHammer identifies vulnerabilities in purification clusters more effectively than traditional methods, significantly enhancing the assessment of diffusion-based purification robustness. Their experiments demonstrate that DiffHammer outperforms existing approaches by identifying more adversarial samples and highlighting previously underestimated security risks.

**Strengths:**

1. This paper is well-sturctrued.

2. They identified the limitations of the N + 1 evaluation for diffusion-based adversarial purification and proposed an EM-based attack and the 1 + N evaluation method, which demonstrated effectiveness.

3. The theoretical foundation of this paper is comprehensive, with detailed proofs provided in the appendix. Extensive experiments further validate the reasonableness of the authors’ method, revealing underestimated risks of diffusion-based adversarial purification.

**Weaknesses:**

1. The paper describes that when the purification process contains unshared vulnerabilities, existing EoT-based N-averaging can encounter the gradient dilemma. This gradient dilemma might lead to the inability to generate effective attack samples, resulting in insufficient evaluation of diffusion-based purification methods. However, the paper lacks theoretical proof or experimental validation for this gradient dilemma.

2. The paper mentions that diffusion model-based purification methods like Diffpure, GDMP, and LM have issues with resisting resubmit attacks. However, to my knowledge, DiffSmooth[1], another diffusion model-based purification method, takes into account multiple denoised outputs for the same input sample when making predictions, and it claims to have a certifiably robust pipeline. This defense method appears to be more powerful, and the authors should consider including such methods when evaluating the robustness of diffusion-based adversarial purification techniques. Additionally, it would be valuable to investigate whether resubmit attacks still pose a problem for such robust defense methods.

[1] Zhang J, Chen Z, Zhang H, et al. {DiffSmooth}: Certifiably robust learning via diffusion models and local smoothing[C]//32nd USENIX Security Symposium (USENIX Security 23). 2023: 4787-4804.

**Questions:**

See weaknesses.

**Limitations:**

The authors have clearly described the limitations.

---

> ### Author Rebuttal · Authors · 2024-08-07
>
> Thank you for the valuable feedback, here's our response to the concerned problems.
> ## Validation of the gradient dilemma
> **Summary:** By conducting cluster analysis on the gradients of the purification process, we observed that the gradients have multiple clustering centers with low inter-cluster similarity, leading to what we term the *"gradient dilemma"* in attacks. This phenomenon is illustrated using a simple Gaussian mixture model, suggesting that the gradient dilemma may stem from non-clustered features in the data.
>
> For our analysis, we sampled 500 gradients for each of the defenses. We employed a Gaussian mixture model (GMM) as the clustering algorithm, consistent with our paper's methodology. The optimal number of clusters is determined using the Akaike Information Criterion (AIC) within the range of 1 to 5. Our experiment revealed that 47% of the sampled gradients (across 3 defenses) possess more than one clustering center as shown in Rebuttal Figure 1. For these samples, we further analyzed the distribution of the cosine similarity between the top two cluster centers as shown in Rebuttal Figure 2, which was often in (-0.2 $\sim$ 0.3), thereby confirming the presence of the gradient dilemma.
>
> To investigate the origins of the gradient dilemma, we constructed a Gaussian mixture toy example where the two diagonal components belong to the same category. In this scenario, diffusion-based purification pulls samples slightly towards the origin before diffusing them back into the data distribution. Given that the optimal classifier is a heteroscedastic classifier aligned with the coordinate axes, a sample's gradient direction is either horizontal or vertical, depending on which cluster the sample is closer to.
>
> We demonstrated the probability of a sample's most frequent gradient direction in Rebuttal Figure 3, where values nearing 50% indicating the presence of a gradient dilemma. Simulation results revealed that samples along the diagonal experience severe gradient dilemmas, which is consistent with intuition. In this toy example, the gradient dilemma arises from a non-clustered data distribution, which imposes divergent gravitational pulls for purification. Consequently, we hypothesize that real-world data distributions may also exhibit non-clustered features. For instance, in the misclassification of a cat as a dog, considering the existence of various dog breeds, it is more efficient and effective to distort the features of one specific breed rather than all breeds collectively.
>
> ## Analysis of DiffSmooth vulnerability
>
> **Summary:** The robustness of DiffSmooth benefits from majority voting, a mechanism that enables certifiable robustness. However, its smoothed classifier remains vulnerable to DiffHammer. Although DiffSmooth is robust to resubmit attacks, this occurs at the expense of significant computational cost.
>
> DiffSmooth employs a smoothing classifier in the inner loop and a certification algorithm in the outer loop. The inner loop's smoothing classifier can be viewed as an aggregation of multiple 1-step DiffPure processes. We evaluated DiffSmooth's performance under the $\ell_2$-norm using a ResNet-110 model trained with Gaussian smoothing, adhering to the paper's parameter settings. As illustrated in Figure 1, DiffSmooth's classifier alone is not sufficiently robust; DiffHammer achieves worst-case robustness of 57.0% and 86.7% (out of 10 evaluations) with perturbation budgets of 0.5 and 1, respectively.
>
> Table 1: Experiment results on DiffSmooth classifier
> | Attack | $r=0.5$ |      | $r=1$ |      |
> | ------ | ------- | ---- | ----- | ---- |
> | ASR    | Avg.    | Wor. | Avg.  | Wor. |
> | BPDA   | 25.6    | 46.9 | 50.1  | 86.9 |
> | PGD    | 34.0    | 53.9 | 53.0  | 89.8 |
> | DA     | 25.7    | 45.7 | 49.4  | 86.1 |
> | DH     | 36.8    | 57.6 | 54.7  | 89.8 |
>
> The outer loop's certification algorithm derives robustness through majority voting, providing a lower bound on DiffSmooth's robustness and resisting resubmit adversarial samples with small attack success rate. Although majority voting suppresses resubmit attacks, it encounters several issues:
>
> 1. Achieving theoretical certification requires a vast number of samples (N = 100,000), limiting its practicality.
> 2. DiffSmooth can falsely certify a class when DiffHammer submits adversarial samples with higher ASR. Our experiments show that DiffSmooth will even falsely certify DiffHammer-generated adversarial samples with a radius of 0.175.

---

### Official Review · Reviewer_HpZi · 2024-07-13

**Soundness:** 3
**Presentation:** 2
**Contribution:** 3
**Rating:** 6
**Confidence:** 3

**Summary:**

This paper reveals two limitations of Eot-based attacks in diffusion-based purification: gradient dilemma and underestimation of resubmit attacks' risk. The authors introduce N-time evaluation to evaluate the risk of resubmitted attacks sufficiently. Then, they propose an EM-based attack to solve the gradient dilemma, which achieves satisfactory results compared to both Eot-based and transfer-based attacks. Finally, this paper guides the future improvement of diffusion-based purification.

**Strengths:**

1. This paper has well originality: a novel N-time evaluation is introduced to sufficiently evaluate diffusion-based purification, and an effective EM-based attack is proposed to overcome the shortcomings of existing attacks.
2. The authors revealed two important obstacles in the evaluation of diffusion-based purification and the causes are clearly explained. Moreover, they propose effective evaluation and attack methods to solve the revealed obstacles.

**Weaknesses:**

In experiments:

- In line 232, the author would better explain why they chose "BPDA [1], PGD [21], and AA [6]" as baselines. The published years of [1], [6], and [21] are 2018, 2020, and 2017, it seems these methods are relatively old works. The authors would better consider a comparison with newer methods, such as "Diffusion models for adversarial purification. ICML 2022, https://arxiv.org/abs/2205.07460".
- In lines 239 and 240, the author would better provide the measuring unit of time.
- The authors would better add horizontal and vertical labels for the figures in their paper, including figures 2, 4, 5, and 6.
- In line 268, "as shown in DiffHammer retains the memory of...", it seems to be "as shown in Figure 4, DiffHammer retains the memory of...". In addition, the author would better explain why the transfer-based attack DTI only appears in Table 1 and is not compared in Figure 5 and Figure 4. In subsection 4.3, the authors also don't analysis DTI.
- In line 253, the author would better explain how is "14%" and "28%" calculated or obtained from the experimental results.

**Questions:**

Please check weaknesses part.

**Limitations:**

I think that the authors have adequately addressed the limitations mentioned in this paper and the potential negative societal impact of their work is controllable.

---

> ### Author Rebuttal · Authors · 2024-08-07
>
> Thank you for the valuable feedback, here's our response to the concerned problems.
> ## Comparison with newer methods
> **Summary:** We adhere to the literature's naming conventions for these methods, but our evaluation incorporates the enhancements proposed in \[1,2\] (2023), upgrading PGD and AA to **SOTA evaluation** with exact gradients. Additionally, we test complementary attacks.
>
> Stochastic and iterative algorithms for diffusion-based purification yield challenging gradients, so a line of works aimed at approximating gradients in attack algorithms. This includes the Adjoint method \[3\] and Joint methods (score/full) \[4\]. \[1,2\] obtained exact gradients of DiffPure and GDMP via computational graph reconstruction, and we further derive exact gradients for the optimization in LM using the Hessian vector product trick. It was shown that approximate gradients lead to inadequate robustness evaluation in \[1,2\]. Therefore, we utilize exact gradient algorithms as a better baseline, with complementary experiments on approximate gradients validating this point. We will clarify our evaluation in the paper.
>
> Method|DiffPure|GDMP|LM
> ---|---|---|---
> Adjoint|41.4/68.0|42.3/59.6|N/A
> Joint (full)|35.7/62.7|37.3/54.9|53.1/73.2
> Joint (score)|22.3/56.4|17.5/43.0|36.0/60.9
> Exact Gradient|46.7/69.0|50.6/63.3|82.0/90.4
> DH|53.8/75.2|57.9/72.5|83.3/91.4
>
> ## Transfer-based attack DTI
>
>
> We include a demonstration and discussion of DTI, a representative transfer-based attack approach. We will compare DTI's performance with other algorithms in Figures 4 and 5 of the paper. DTI performs well in transfer-based attacks, likely due to its emphasis on translation and transformation (scaling and affine) invariance. Since most stochastic purifications do not share vulnerabilities, assuming global optimality can invalidate some transfer-based algorithms (VMI and CWA). Whereas DTI does not rely on this assumption, but it only considers simple transformations such as translation and scaling, which limits its performance.
>
> ## Typos and presentation issues
>
> - Since the time spent per step is comparable for the different methods, we directly report the number of iterations required to reach 90% of the optimal metric. We will unify the statements as iterations rather than time to avoid ambiguity.
> - We define the robustness of the defense as the proportion of data that is not misclassified in 10 resubmit attacks, which is 1-Wor.ASR (the proportion of attacks that succeed at least once).
> - We will add horizontal and vertical axis labels to the image to enhance legibility.
> - We will check and correct typos in the paper.
>
> \[1\] Robust Evaluation of Diffusion-Based Adversarial Purification, ICCV, 2023.
>
> \[2\] DiffAttack: Evasion Attacks Against Diffusion-Based Adversarial Purification, NeurIPS, 2023.
>
> \[3\] Diffusion models for adversarial purification, ICML, 2022.
>
> \[4\] Adversarial purification with score-based generative models, ICML, 2021.

---

> > ### Comment · Reviewer_HpZi · 2024-08-09
> >
> > Thank you for your responses.  I think it solves most of my concerns, and I will keep my score.

---

> ### Author Response · Authors · 2024-08-12
>
> We sincerely appreciate your acknowledgment and your positive feedback on our work!

---

### Author Rebuttal · Authors · 2024-08-07

Thank you for the valuable feedback, here's our response to the concerned problems. We have provided some additional images attached to the pdf.

---

### Decision · Program_Chairs · 2024-09-25

**Decision:**

Accept (poster)

**Comment:**

The authors propose a sufficient and efficient attack named DiffHammer against diffusion-based purification.
The reviewers agree that the paper makes interesting analysis and conclusions and please make sure to incorporate the improvements in the final version of the paper/